# New particle formation leads to enhanced cloud condensation nuclei concentrations in Antarctic Peninsula

Jiyeon Park[1], Hyojin Kang[1,2], Yeontae Gim[1], Eunho Jang[1,2], Ki-Tae Park[1], Sangjong Park[1], Chang Hoon Jung[3], Darius Ceburnis[4], Colin O'Dowd[4], and Young Jun Yoon[1, *]

[1]Korea Polar Research Institute, 26 Songdomirae-ro, Yeonsu-gu, Incheon 21990, South Korea
[2]University of Science and Technology (UST), 217 Gajeong-ro, Yuseong-gu, Daejeon, Republic of Korea
[3]Department of Health Management, Kyungin Women's University, Incheon 21041, Republic of Korea
[4]School of Natural Sciences and Centre for Climate and Air Pollution Studies, Ryan Institute, University of Galway, Ireland
*Correspondence to: Y.J. Yoon (yjyoon@kopri.re.kr)

## Abstract

Few studies have investigated the impact of new particle formation (NPF) on cloud condensation nuclei (CCN) in remote Antarctica, and none has elucidated the relationship between NPF and CCN production. To address that knowledge gap, we continuously measured the number size distribution of 2.5–300 nm particles and CCN number concentrations at King Sejong Station in the Antarctic Peninsula from January 1 to December 31, 2018. Ninety-seven new particle formation (NPF) events were detected throughout the year. Clear annual and seasonal patterns of NPF were observed: high concentration and frequency of nucleation-mode particles in summer (December–February: 53 NPF cases) and undetected nucleation-mode particles in winter (June–August: no NPF cases). We estimated the spatial scale of NPF by multiplying the time during which a distinct nucleation mode can be observed at the sampling site by the locally measured wind speed. The estimated median spatial scale of NPF around Antarctic peninsula was found to be approximately 155 km, indicating the large-scale of NPF events. Air back-trajectory analysis revealed that 80 cases of NPF events were associated with air masses originating over the ocean, followed by sea-ice (12 cases), multiple (3 cases), and land (2 cases) regions. We present and discuss three major NPF categories: (1) marine NPF (2) sea-ice NPF, and (3) multiple NPF. Satellite-estimates for sea surface dimethylsulfoniopropionate (DMSP; a precursor of gaseous dimethyl sulfide) data showed

that the production of oceanic biogenic precursors could be a key component in marine NPF events, whereas halogen compounds released from ice-covered areas could contribute to sea-ice NPF events. Terrestrial sources (wild life colonies, vegetation, and meltwater ponds) from Antarctica could affect aerosol production in multiple air masses. Out of 97 observed NPF events, 83 cases were characterized by the simultaneous increase in the CCN concentration by 2–270% (median 44%) in the following 1 to 36 hours (median 8 hours) after NPF events. Overall, Antarctic NPF events were found to be a significant source of particles with different physical characteristics and related to biogenic sources in and around the Antarctic Peninsula, which subsequently grew to cloud condensation nuclei.

## 1. Introduction

Antarctic peninsula is warming more rapidly than Earth's global mean rate (Chen et al., 2009; Vaughan et al., 2003), leading to shrinking sea-ice coverage and consequent sea-level rise (Pritchard et al., 2009). In the Antarctic region, ambient aerosols play a crucial role in governing radiative transfer, directly by the scattering and absorption of solar radiation and indirectly by acting as cloud condensation nuclei (CCN) (IPCC, 2013). The magnitude of the radiative forcing caused by the interactions between aerosols and CCN remains highly uncertain due to a poor understanding of pristine natural aerosols (Carslaw et al., 2013). To reduce this uncertainty, the physicochemical properties of aerosol particles (e.g., number concentrations, size distributions, chemical compositions, and hygroscopicity) have been studied at several Antarctic stations including King Sejong Station (Kim et al., 2019), Aboa (Asmi et al., 2010; Virkkula et al., 2006), Dome C (Järvinen et al., 2013), Halley (Lachlan-Cope et al., 2020; O'Dowd et al., 1997), Kohen (Weller et al., 2018), McMurdo (Giordano et al., 2018; Liu et al., 2018), Neumayer (Teinilä et al., 2014; Weller et al., 2015), Princess Elisabeth (Herenz et al., 2019) and Syowa (Hara et al., 2011; Ito, 1993). Furthermore, open ocean and coastal Antarctic expeditions such as SIPEXII (Sea Ice Physics and Ecosystems eXperiment, 2012; Humphries et al., 2015; Humphries et al., 2016), PEGASO (Plankton-derived Emissions of trace Gases and Aerosols in the Southern Ocean, 2015; Dall'Osto et al., 2017; Decesari et al., 2020; Fossum et al., 2018), ACE-SPACE (Antarctic Circumnavigation Expedition – Study

of Preindustrial-like Aerosol Climate Effects, 2017; Schmale et al., 2019; Walton and Thomas, 2018), PCAN (Polar Cell Aerosol Nucleation, 2017; Simmons et al., 2021); PI-ICE (Polar atmosphere-ice-ocean Interactions: Impact on Climate and Ecology, 2019; Brean et al., 2021; Dall'Osto et al., 2022) studies on the influences of marine aerosols on climate and ecology. Overall, aerosol particle number concentrations follow a clear annual trend, being much higher in austral summer than in other seasons (Järvinen et al., 2013; Kerminen et al., 2018; Weller et al., 2011). For instance, Kim et al. (2017) found that summertime concentrations in the Antarctic Peninsula were ~20 times higher than in winter. This pattern can be largely explained by new particle formation (NPF) events.

Precursor gases for NPF in this region can originate from the ocean, sea-ice, meltwater ponds, terrestrial animal colonies, anthropogenic activity and continental ecosystem. Oceanic emissions of dimethyl sulfide (DMS) represent the largest natural sulfur source in the Antarctic atmosphere (Simó, 2001), and its photooxidation is a key process contributing to NPF (Giordano et al., 2017; Jang et al., 2019 and 2022). For instance, in situ (Saiz-Lopez et al., 2007) and satellite (Schönhardt et al., 2008) measurements have shown Antarctica to be an iodine emission hotspot, particularly from the sea-ice in the Weddell Sea during spring (Atkinson et al., 2012). Indeed, Sipilä et al. (2016) measured iodic acid ($HIO_3$) in Antarctica and found that the Antarctic oceanic regions may be strong sources of molecular iodine, which is then converted to $HIO_3$ in gas-phase reactions. Dall´Osto et al. (2017) reported that microbiota in sea-ice were associated with atmospheric organic nitrogen formation in the Southern Ocean near Antarctica. According to Kyrö et al. (2013), the precursor vapors responsible for NPF and subsequent growth could originate from the cyanobacteria, which are abundant in Antarctic meltwater ponds. In addition, continental Antarctica is a habitat for various types of seabirds and penguins, with guano species acting as a crucial source of ammonia and organic compounds and may contribute to NPF in coastal Antarctic areas (Schmale et al., 2013; Weber et al., 1998; Zhu et al., 2011). At continental South Pole NPF event are commonly associated with the local anthropogenic pollution during calm weather conditions (Park et al., 2004). In addition, the biomass burning aerosol from South American continental outflow has been observed at Troll Research Station (Fiebig et al., 2009). During the daytime, higher radiation

enhances photo-active emissions from land ecosystems (mosses, grasses, and lichens) of the Antarctic Peninsula and can lead to NPF and aerosol growth (Decesari et al., 2020; Quéléver et al., 2022; Schmale et al., 2013). However, land sources are rather unlikely due to a small footprint of emerging land and the associated short overpass over the sparse vegetation.

In recent years, long-term records of aerosol size distribution have become an important aspect of investigations into the sources and dynamical processes of NPF. The majority of Antarctic field studies have focused on the annual and spatial patterns of the number size distribution of particles > 10 nm (Belosi et al., 2012; Järvinen et al., 2013; Kim et al., 2019; Kyrö et al., 2013; Lachlan-Cope et al., 2020). Although NPF events are typically characterized by a rapid increase in the number concentration of cluster from 1–3 nm (Kulmala et al., 2004), datasets for these types of aerosol size distribution remain rare. To date, number size distribution of particles > 3 nm has been reported by Asmi et al. (2010) at Aboa during from December 29, 2006 to January 29, 2007; by Pant at al. (2011) at Maitri from January 1 to February 28, 2015; by Weller et al. (2015) at Neumayer from January 20 to March 26, 2012; by Jokinen et al. (2018) at Aboa from November 2014 to February 2015; by Weller et al. (2018) at Kohnen during January 2015 and 2016; by Quéléver et al. (2022) at Marambio during the austral summer between January 15 and February 25, 2018; and by Brean et al. (2021) during the PI-ICE cruise from January 25 to February 4, 2019. However, all of these measurements were made during the Antarctic summer due to restricted access and, therefore, limited information on seasonal cycles.

Newly formed particles can grow into larger sizes that act as CCN, becoming relevant for cloud formation (O'Dowd, 2002; Williamson et al., 2019). In a highly pristine atmosphere such as Antarctica, where CCN concentration is extremely low (Kim et al., 2017), NPF may be a significant phenomenon controlling the CCN budget (Kyrö et al., 2013). For instance, Herenz et al. (2019) showed that an elevated $CN_{2.5}$ (total number concentration of particles > 2.5 nm) during NPF events was accompanied by an increase in CCN concentrations at Princess Elisabeth during austral summer (December to February, 2013–2016). Ship-based observations during the ACE-SPACE found that the fraction of particle serving as CCN was higher near the coast of Antarctica compared to open ocean, resulting from multiple

processing cycles of dissipating and condensing clouds and/or the higher availability condensable gases

originating from marine microbial activity (Schmale et al., 2019). In addition, seasonal variability in

$CN_{2.5-10}$ (number concentration of particles within the 2.5 nm and 10 nm range and attributed to NPF)

and CCN concentrations at King Sejong Station from March 2009 to December 2016 were investigated

by Kim et al. (2019), who concluded that CCN concentrations during NPF events increased by ~11%

compared to the background concentration. However, to date, only one study (Kim et al., 2019) has

reported the contribution of NPF to CCN in the Antarctic Peninsula, and that study did not consider

aerosol number size distribution.

In this study, we continuously recorded the number size distribution of 2.5–300 nm particles and

CCN number concentrations at King Sejong Station in the Antarctic Peninsula from January 1, 2018, to

December 31, 2018. Our primary goals were to (1) characterize the seasonal variation and occurrence of

NPF events from the perspective of aerosol physical properties (total number concentration, number size

distribution, formation and growth rates, and condensation sink); (2) improve our understanding of the

major sources (including open ocean, sea-ice, and land) and processes influencing NPF and particle

growth; and (3) estimate the contribution of atmospheric NPF to CCN activity in this pristine environment.

To our knowledge, this is the first study to present direct evidence of CCN production associated with

NPF and growth events in the Antarctic Peninsula, using simultaneous measurements of particle number

size distributions (down to 3 nm) and CCN properties for a full year.

**2. Experimental methods**

**2.1. Sampling site and instrumentation**

Continuous measurements of the physical properties of aerosol particles were conducted from

January 2018 to December 2018 at King Sejong Station in the Antarctic Peninsula (62.22° S, 58.78° W).

Full details of the sampling site and measurement setup are given in Kim et al. (2017). In brief, a

cylindrical stainless inlet (0.1 m diameter and 5.2 m length; total flow rate of the sampled air was 150 L

133 min$^{-1}$) was placed on the observatory roof following Global Atmosphere Watch aerosol measurement

guidelines and recommendations. Two condensation particle counters (TSI model 3776 CPC and TSI model 3772 CPC) were used to measure the total number concentration of particles larger than 2.5 (corresponding data $CN_{2.5}$) and 10 nm (corresponding data $CN_{10}$) every 1 s, respectively. The aerosol sample flow rates of TSI model 3776 CPC and TSI model 3772 CPC were 1.5 and 1.0 L min$^{-1}$, respectively. A nano-scanning mobility particle sizer (nano-SMPS) consisting of a nano-differential mobility analyzer (nano-DMA) (TSI model 3085, USA) and an ultrafine condensation particle counter (TSI model 3776, USA) was used to measure the number size distribution of particles from 2.5–64 nm every 3 minutes. The aerosol flow rate was 1.5 L min$^{-1}$ and the sheath flow rate was 15 L min$^{-1}$ inside the nano-DMA.

The particle number size distribution (from 10–300 nm every 3 min) was measured with a standard-SMPS consisting of a long DMA (TSI model 3081, USA) and a CPC (TSI model 3772, USA). The aerosol flow rate was 1.0 L min$^{-1}$, and the sheath flow rate was 10 L min$^{-1}$ inside the long DMA. To obtain the number size distribution of particles from 2.5–300 nm, the nano-SMPS and standard-SMPS were merged. For particle diameters 2.5–20 nm, nano-SMPS data were chosen because this was optimized to operate with a smaller particle diameter. In the nano-DMA, the aerosol residence time can be reduced by shortening the inlet transport passage (5.0 cm) and increasing the inlet flow (up to 16.5 L min$^{-1}$) ($< 10$ nm) (Chen et al., 1998). Hence, the number size distribution data from both nano-SMPS and standard-SMPS were merged at a diameter of 20 nm. Furthermore, three-point median filter and five point moving average were performed on merging the number size distribution data to remove nano-SMPS noise, as suggested by Kulmala et al. (2012).

The black carbon (BC) concentration was measured using an aethalometer (AE22, Magee Scientific Co., USA) every 5 min to examine long-range polluted aerosol transport from other continents and to assess the influence of local pollution from the station. The flow rate through a sharp-cut 2.5 μm cyclone (BGI, Inc., USA) was set to 5 L min$^{-1}$. The CCN counter (CCNC: CCN-100, Droplet Measurement Technologies, USA) measured CCN number concentrations at five different supersaturation levels of 0.2% 0.4%, 0.6%, 0.8%, and 1% every 30 minute. The total flow rate in the CCN counter was 0.5 L min$^{-1}$. The sample and sheath flow rates of the CCN counter were 0.05 and 0.45 L min$^{-1}$, respectively. In addition,

basic meteorological parameters (temperature, pressure, relative humidity (RH), wind speed, wind direction, and solar radiation intensity) were measured using an automatic weather station (Vaisala HMP45).

**2.2. Data evaluation**

As the observatory is located ~400 m southwest of the main station buildings and several kilometers away from other research stations, measurement data were impacted by local emissions from station activities (e.g., power generators and incineration) or anthropogenic pollutions near the observatory (e.g., plumes from other research station about several kilometers, vessels providing research station supply, and commercial cruise vessels). To obtain an unperturbed aerosol population of pristine Antarctic environment, contaminated measurements were removed manually based on wind direction, wind speed, BC concentration, and total particle number concentration. The following data elimination procedure was applied: (1) the measurements taken within wind sector of 355° and 55° were discarded as directly impacted by local pollution sources; (2) relative wind speed below 2.0 m s$^{-1}$, as stagnant conditions would have facilitated contaminated particle propagation to the measurement location; (3) equivalent BC mass concentrations exceeding 50 ng m$^{-3}$, because elevated BC concentration unambiguously pointed at polluted particles; and (4) a sharp increase in the total number concentration over the entire particle diameter range in a short time scale of less than an hour, as such abrupt peaks and spikes are related to potential contamination or instrumental malfunctions. For instance, CPC and SMPS data were removed for time periods when particle number concentrations suddenly increased to more than twice the background values.

Based on a four-year (2016-2019) BC dataset, six types of Antarctic Peninsula air-pollution levels were identified (Grigas et al., 2017): (1) pristine air with BC concentrations < 15 ng m$^{-3}$, (2) clean air with BC levels 15–50 ng m$^{-3}$, (3) slightly polluted air with BC levels 50–100 ng m$^{-3}$, (4) moderately polluted air with BC levels 100–300 ng m$^{-3}$, (5) polluted air with BC levels 300–1000 ng m$^{-3}$, and (6) extremely polluted air with BC concentrations > 1000 ng m$^{-3}$ (Figure 1). Previously, BC data were used as indicators

for local contamination in Antarctica when BC concentration level exceeded 50 ng m$^{-3}$ (Herenz et al., 2019) or 100 ng m$^{-3}$ (Jang et al., 2018; Kim et al., 2017; Kim et al., 2019; Weller et al., 2011; Weller et al., 2015). Hara et al. (2019) measured BC concentration at Syowa station Antarctica from February 2005 until December 2016. They found that the daily median BC concentrations were below the detection limit (0.2 ng m$^{-3}$) to 63.8 ng m$^{-3}$ at Syowa Station (median, 1.8 ng m$^{-3}$; mean, 2.7 ng m$^{-3}$ during the measurement period). During the ACE-SPACE expedition, BC concentration reach its background levels of 19.2 ng m$^{-3}$ (Schmale et al., 2019). Arctic shipborne-observations measured BC concentration throughout the Arctic Ocean and Pacific Ocean during the summer of 2017, all pointing to pristine clean marine air masses with BC values of approximately $20 \pm 10$ ng m$^{-3}$. (Park et al., 2020).

Of the total time period assessed, pristine air conditions represented 30% (mean value of BC: $6 \pm 6$ ng m$^{-3}$), clean for 44% (mean value of BC: $30 \pm 10$ ng m$^{-3}$), lightly polluted 19% (mean value of BC: $69 \pm 14$ ng m$^{-3}$), moderately polluted 6% (mean value of BC: $150 \pm 47$ ng m$^{-3}$), polluted 1% (mean value of BC: $499 \pm 174$ ng m$^{-3}$), and extremely polluted less than 1% (mean value of BC: $1537 \pm 595$ ng m$^{-3}$). Together, pristine and clean air conditions accounted for ~74% of the time with the remaining 26% (BC > 50 ng m$^{-3}$) removed prior to data analysis.

## 2.3. Definition of NPF and growth events

NPF events were visually identified by the particle number size distribution based on the protocol described by Dal Maso et al. (2005) and Kulmala et al. (2012). Here, these were defined when a distinct new mode of particles (initially < 25 nm), appearing in the particle number size distribution at nucleation-mode size (3−25 nm), prevailed for more than an hour. Using these criteria, the particle size distribution data showed that in some cases, there was only a short burst of nucleation-mode particles without clearly discernible particle growth, whereas in other cases, particle formation with subsequent particle growth lasted for several hours, representing a regional-scale phenomenon (Ström et al., 2009). This enabled us to determine the particle growth rate (GR), which is not possible during short bursts of nucleation-mode particles.

The particle growth and formation rates along with the condensation sink were calculated from the

measured particle number size distribution. The GR was determined using the maximum concentration

and mode-fitting methods (Dal Maso et al., 2005; Yli-Juuti et al., 2009). GR was calculated by a linear fit

through the geometric mean diameter of the nucleation-mode particles as a function of time during NPF.

The formation rate (FR) of nucleation-mode particles ($J_{3\text{-}25}$) was calculated by taking into account the

time evolution of the particle number concentration in this size range and particle losses due to the

coagulation sink and condensational growth out of the size range (Kulmala et al., 2012). The surface area

of particles available for the condensation of gaseous molecules can be characterized by a condensation

sink (CS), which determines how rapidly vapor molecules condense onto pre-existing particles (Collins

et al., 2017; Dal Maso et al., 2002).

**2.4. Backward trajectory analysis and potential source regions**

Air mass back trajectories were obtained using the Hybrid Single-Particle Lagrangian Integrated

Trajectory (HYSPLIT) model to investigate their relationships with the physical characteristics of aerosol

particles (Draxler and Hess, 1998). The 2 days air mass back trajectories (48 hours) were determined at

hourly intervals and combined with satellite-derived geographical information to estimate the transport

history of the air masses arriving at the observation site (Jang et al., 2022 and Park et al., 2021). The

potential origins of the aerosols were divided into three categories based on the retention time of the 2

230    days back trajectories over three major domains: ocean (including the Weddell and Bellingshausen Seas),

sea-ice, and land (including the Antarctic Peninsula). Daily geographical information on ocean, sea-ice,

and land area was obtained from the Sea Ice Index (25 km resolution) provided by the National Snow and

Ice Data Center (NSIDC). The sea-ice zone was defined as the area with a sea-ice coverage >15% (Stroeve

et al., 2016). Air masses that passed over the Weddell and Bellingshausen Sea-regions were categorized

as originating from the ocean (i.e. > 50% retention over the ocean region). The air masses that frequently

advected over the sea-ice region were categorized as originated over the sea-ice (i.e. > 50% retention over

the sea-ice domain). Air masses that traveled through the Antarctic Peninsula were categorized as

originating from the land (i.e. > 50% retention over the land). Finally, the air masses which passed over the ocean, sea-ice, and land regions simultaneously were categorized as originating from the multiple regions (i.e., 20–40 % retention over each ocean, sea-ice, and land domain).

To evaluate the influence of oceanic biological characteristics on NPF properties, the phytoplankton biomass of the ocean domains was estimated by calculating their chlorophyll concentration from the Moderate Resolution Imaging Spectroradiometer on the Aqua (MODIS-Aqua) satellite at 4 km resolution during the entire study period. Phytoplankton produces dimethylsulfoniopropionate (DMSP, a precursor of gaseous DMS) and other organic vapors all of which are potential precursors to new particle formation. Thus, the spatiotemporal distribution of sea-surface DMSP could be an indicator of contemporary DMS emissions. The total DMSP concentration on the sea-surface was estimated using the algorithm developed by Galí et al. (2015). The algorithm for the total DMSP concentration was based on the satellite-derived chlorophyll concentration and photosynthetic radiation exposure. To calculate the air mass exposures to ocean chlorophyll and DMSP (Jang et al., 2019), hourly back trajectory position was combined with satellite-derived chlorophyll concentration and total DMSP concentration, providing a good measure for quantitatively investigating the biological exposure history of sampled air over the several days before its arrival at the observation site (Park et al., 2018 and 2021). However, satellites are not typically used to directly measure the biomass of sea ice algae because satellite sensors cannot penetrate through thick layers of sea ice to directly measure the biomass of algae beneath it (Lee et al., 2015; Lange et al., 2017). Thus, calculated chlorophyll exposures (i.e., satellite-estimates of biological activity) cannot account for the biological activities thriving within and beneath of the sea ice.

## 3. Results and discussion

### 3.1. General features and annual cycle

We investigated the overall seasonality of particle number size distributions focusing on NPF events. In addition, local meteorological parameters (e.g., temperature, RH, wind speed, wind direction, pressure, and solar radiation) and air mass back trajectories were used to support the interpretation of the seasonal

trends of the particle number size distribution and the dynamics of NPF events observed at the station.

### 3.1.1. Particle number concentrations and size distributions

Figure 2 shows a time series of the one-hour average total particle number concentration and size-
segregated particle number concentrations over the entire measurement period conforming to pristine (BC
$< 15$ ng m$^{-3}$) and clean (BC: 15–50 ng m$^{-3}$) conditions. In addition, monthly medians for total number
concentration of particles, size-segregated particles number concentration, CCN number concentration at
supersaturation of 0.4%, and meteorological parameters are included in Table 1. The $CN_{2.5}$ and $CN_{10}$
ranged from 60 to 3982 cm$^{-3}$ and 30 to 3304 cm$^{-3}$, respectively. The annual median number concentrations
of particles for the nucleation mode ($N_{NUC}$; 2.5–25 nm in diameter), Aitken mode ($N_{AIT}$; 25–100 nm in
diameter), and accumulation mode ($N_{ACC}$; 100–300 nm in diameter) were 46.8 cm$^{-3}$, 53.5 cm$^{-3}$, and 21.7
274 cm$^{-3}$, respectively. The highest median $N_{NUC}$, $N_{AIT}$, and $N_{ACC}$ values were recorded in December (193.5
275 cm$^{-3}$), December (227.6 cm$^{-3}$), and January (83.8 cm$^{-3}$), respectively (Table 1). The lowest $N_{NUC}$, $N_{AIT}$,
and $N_{ACC}$ values were recorded during austral winter in June – 12.2 cm$^{-3}$, 12.5 cm$^{-3}$ and 9.2 cm$^{-3}$,
respectively. Overall, clear annual and seasonal patterns of particle number concentrations in all size
classes were observed: high concentrations in summer (December–February) and low concentrations in
winter (June–August), similar to those observed at Marambio Station in the Antarctic Peninsula (Asmi et
al, 2018), at coastal Neumayer Station (Weller et al., 2011), at Concordia Station Dome C (Järvinen et al.,
2013), and at Troll Station (Fiebig et al., 2014). Furthermore, the hourly average $CN_{10}$ value was
positively correlated with the hourly average $N_{NUC}$ ($R = 0.88$; not shown), implying that the summer
maximum of total particle number concentrations was largely influenced by newly formed particles in the
Antarctic atmosphere.

### 3.1.2. Influence of meteorological parameters on NPF events

The meteorological parameters after data filtering (BC $< 50$ ng m$^{-3}$ indicating pristine and clean
conditions) were characterized by a solar radiation range of 0–919 W m$^{-2}$ (median 10.7 W m$^{-2}$), a
temperature range of -20–6 °C (median -1.2 °C), an RH range of 52–98 % (median 88 %), a pressure
range of 950–1022 hPa (median 988 hPa), a wind speed range of 0.3–21 m s$^{-1}$ (median 7.4 m s$^{-1}$), and

wind direction range of 3–357° (median 296°) (Figure S1). To understand impacts on the particle number size distributions, we determined the relationships between the size-segregated particle number concentrations and meteorological parameters (Figure S2). $CN_{10}$, $N_{NUC}$, $N_{AIT}$, and $N_{ACC}$ were positively correlated with both solar radiation intensity and temperature. In particular, $N_{NUC}$ had the highest correlation with solar radiation intensity ($R = 0.39$) of any meteorological condition, suggesting that solar radiation is one of the most important factors influencing NPF events, as it can drive photochemical reactions leading to the production and further reaction of precursor gases. In contrast, there was a weak anticorrelation between RH and $N_{NUC}$, supporting the view that NPF occurs preferentially at low RH (Dada et al., 2017; Hamed et al., 2011; Jeong et al., 2010; Laaksonen et al., 2008). Field observations have reported that during NPF events, RH was negatively related to the number concentration of freshly formed particles (Jeong et al., 2004; Lachlan-Cope et al., 2020; Weber et al., 1997) because of the enhanced coagulation from scavenging effect of sub-3 nm nanoparticles at high RH and the diminished solar radiation at high RH. Previously, some NPF events were associated with high wind speeds at various Antarctic stations, such as Neumayer (Weller et al., 2015) and Aboa (Asmi et al., 2010; Virkkula et al., 2007). These studies found an enhanced particle number concentration < 10 nm during stormy weather and suggested ion production by frictional processes in fast-moving snow and ice crystals, followed by subsequent ion-mediated nucleation during strong winds. However, in our study, wind speed was not correlated with $N_{NUC}$ ($R = -0.18$), $N_{AIT}$ ($R = -0.04$), or $N_{ACC}$ ($R = -0.05$), as recently suggested by Liu et al. (2018). Our results indicated that wind speed did not affect NPF events. A possible explanation for the wind speed independence is that an increase in wind speed contributes to the increase of cluster size ion number concentrations by friction processes (Virkkula et al., 2007), but it was also accompanied by cloudy conditions. In summary, the elevated $N_{NUC}$ values (i.e., indicator of NPF events) at King Sejong Station were more likely to be accompanied by high solar radiation, high temperature, and low RH, regardless of wind speed. Recent studies concluded that Antarctic NPF occurred under combined high solar radiation, high temperature and low RH conditions, similar to previous study measured at the Marambio Antarctic research station (Quéléver et al., 2022). Quéléver et al. (2022) found all NPF events

were observed during the daytime with high solar radiation (clear-sky conditions), mostly with above-freezing temperature and with low RH.

### 3.1.3. Characteristics of NPF events

NPF events in this study were identified based on the size distribution data measured using the standard-SMPS (Figure 2e) and nano-SMPS (Figure 2f). During the pristine and clean periods (comprising of 355 observation days and 169166 size distribution spectra for the standard-SMPS, and of 349 observation days and 165259 size distribution spectra for nano-SMPS), NPF events were frequently observed at King Sejong Station, as shown by the size distribution data (Figure 2f). 97 events (26% of observation days) with elevated $N_{NUC}$ were observed when taking place in pristine (BC < 15 ng m$^{-3}$) and clean (BC: 15–50 ng m$^{-3}$) conditions. Median value of BC concentrations during NPF events was 21.0 ng m$^{-3}$, similar to that of whole measurement periods after data filtering (median BC value: 18.8 ng m$^{-3}$) (section 2.2). This indicated that NPF events are independent of occasional increases of BC during clean periods. The NPF events were classified into: (1) burst event and (2) nucleation with growth event according to the classification by Dal Maso et al. (2005) as seen in Figure S3. The burst events and nucleation with growth events were observed on 1 January 2018 and 16 December 2018, respectively. NPF events were more frequently observed in summer (~55%) than in any other season (Figure 4), with the highest frequency in January (22%) and December (22%) followed by spring (September–November, 34%) and autumn (March–May, 11%). Similar results were reported by Järvinen et al. (2013) based on observations from Dome C and Kim et al. (2019) based on observations from King Sejong Station. Although Järvinen et al. (2013) reported winter events that occurred in the absence of sunlight, we did not detect NPF events during austral winter from May through to August.

In order to investigate the seasonal characteristics of NPF event, we compared mean size distributions of aerosol particles for summer, spring (transition period of the melting ocean), and autumn (transition period of refreezing of the ocean) (Figure S4). Trimodal distributions were presented in all seasons excepting winter when nucleation mode or particle formation was not observed. For instance, a trimodal distribution was seen at 7 nm, 30 nm, 122 nm during summer months. The number concentration

of nucleation and Aitken modes were higher than the accumulation modes, indicating that NPF event regulates the aerosol processes in Antarctic peninsula. The largest mode at 126 nm or 103 nm may be due to a combination of primary (produced by bubble-bursting process) and secondary (produced by gas-to-particle conversion process) aerosol components. Results are broadly in line with previous results published from the Arctic and Antarctic regions. A ship-borne field campaign over Arctic Ocean found a trimodal distribution at 18 nm, 53 nm and 150 nm for open-ocean marine Arctic NPF event (Park et al., 2020). Lachlan-Cop et al. (2020) presented k-mean cluster analysis of particle size distribution measured at Halley, Antarctica, showing a nucleation peak at 15 nm for "nucleation" ultrafine category and a nucleation peak at 27 nm for "bursting" ultrafine category.

Air mass back trajectories were calculated at hourly intervals to investigate possible source regions for the observed NPF events. Figure 2g shows the residence times of air masses over the three domains. Based on 2-days air mass transport history analysis, air masses allocated to ocean, sea-ice and land account for 83, 12 and 5%, respectively, during the study period. 97 cases were identified as NPF events, 80 of which were observed when the air mass originated over the ocean domain (Figure 4). 12 NPF events were observed in air masses originating over the sea-ice domain, while the remaining 5 events were associated with multi-regional origin (3 cases) and land origin (2 cases). Multi-regional origin indicated air masses simultaneously influenced by all three domains. Median BC concentration for ocean, sea-ice, and multiple air masses found to be 23.8 ng m$^{-3}$, 12.7 ng m$^{-3}$, 9.8 ng m$^{-3}$, respectively, (Figure S5), indicating pristine clean air masses with minimum influence from anthropogenic pollutions during each NPF event case. Our results indicated that NPF events were more common in air masses originating over the ocean and sea-ice compared to those originating from the land. Precursors released by both ocean and sea-ice could play an important role in the formation of new particles in the Antarctic atmosphere.

**3.1.4. Spatial extension of regional nucleation event**

Many previous studies have reported that key steps of the nucleation process (e.g., cluster stabilization) occur in the size range ~2 nm, in line with recent direct observations of atmospheric molecular clusters (Kerminen et al., 2018; Kulmala et al., 2013). However, during NPF events, we did

not observe particle formation starting directly from the lower end of the particle size spectrum (2.5 nm), showing that the formation of freshly nucleated particles could not have actually taken place at the site. Indeed, the initial diameter of particles that arrived to the measurement site during the NPF ranged from 4 nm to 16 nm (Figure 3a). Median values of NPF event duration (Figure 3b) and growth rate (Figure 3c) were 4.0 hour and 0.83 nm hr$^{-1}$, respectively. We assumed that they were transported from elsewhere or produced aloft, and detected the appearance of an already grown mode. Consistent with these studies, NPF events can be a regional-scale phenomenon extending over spatial scales of tens to hundreds of kilometers in several regions, such as the remote marine boundary layer (Zheng et al., 2021), Canadian high Arctic (Eureka, Nunavut, on Ellesmere Island in the Canadian Arctic Archipelago) (Tremblay et al., 2019), and Arctic ship-based observations.

Assuming the region is characterized by homogenous meteorological conditions, we estimated the spatial scale of NPF by multiplying the time during which a distinct nucleation mode can be observed at the sampling site by the locally measured wind speed (Birmili et al., 2003; Crippa and Pryor, 2013). As shown in Figure 3d, the spatial extend of NPF event associated with substantial particle growth can be 16–816 km (median value: 155 km), indicating the large-scale NPF events. Weller et al. (2015) measured size distributions at the coastal Antarctic station Neumayer during two summer campaigns (from 20 January to 26 March 2012 and 1 February to 30 April 2014). They found that the spatial extend of NPF event was estimated to be around $170 \pm 85$ km, taking into account the prevailing wind velocity (around $8 \pm 4$ m s$^{-1}$) and the confined NPF duration (around 6h).

## 3.2. Case studies

This section presents a detailed overview of the ocean, sea ice, and multi-regional NPF events.

### 3.2.1. Marine NPF event

A striking series of NPF events took place over seven days (Figure 6), starting at approximately 00:00 on December 9, 2018. Events starting at midnight can likely indicate their formation few hours earlier during afternoon sunlight, because the events are observed with an already grown nucleation mode.

Time series of meteorological parameters, air mass origins, oceanic biological activity (estimated by chlorophyll and DMSP exposures), particle size distribution (measured by nano-SMPS and standard-SMPS), and CCN concentrations are shown in Figure 6. During this time, the prevailing northerly winds (median 307 °) were stable at 7.7 m s$^{-1}$. Air temperature varied from -1.5 to 2.1 °C (median 0.5 °C) and RH varied from 75–97% (median 89 %). There were no data for solar radiation during these events. Air masses predominantly traveled over the Antarctic Ocean (46.9, 0.7, and 0.4 h over ocean, land, and sea ice, respectively) and could be categorized as originating from the Antarctic Ocean. Specifically, the air mass originated mainly from Bellingshausen Sea (Figure 5a). During this event, the median total DMSP and chlorophyll exposures in the surface sea were 18 nmol L$^{-1}$ and 0.26 mg m$^{-3}$, respectively.

Between 00:00 and 20:00 on December 9, $N_{NUC}$ increased from 196 to 688 cm$^{-3}$. At the same time, CCN concentrations at 0.4 % supersaturation gradually increased from 138 (00:00 on December 0) to 326 cm$^{-3}$ (12:00 on December 11), an increase of 135%. In addition, elevated $N_{NUC}$ occurred at 00:00 on December 13, ranging from 118–522 cm$^{-3}$. CCN number concentration at 0.4% supersaturation began to increase at this time (95 cm$^{-3}$) and reached its maximum at 18:00 (503 cm$^{-3}$), with a concentration increase of 431%.

**3.2.2. Sea-ice NPF event**

The NPF event with subsequent particle growth were detected from around 19:00 on January 13, 2018, to around 08:00 on January 14, 2018 (Figure 7). Air temperature and RH during the event were 0.1 °C and 85%, respectively, while solar radiation decreased from 131.7 to 0.2 W m$^{-2}$. Winds were mild and stable (1.9–5.7 m s$^{-1}$), with a prevailing northwesterly (262–350°) direction and air masses predominantly coming from sea-ice. The average retention times of the 2 d back trajectories traveling over ocean, sea-ice, and land were 20.0, 20.9, and 7.1 h, respectively, indicating sea-ice-influenced air masses (Figure 5b). During the NPF event, both total DMSP and chlorophyll exposure values are stable, with median exposures of 13.3 nmol L$^{-1}$ and 0.2 mg m$^{-3}$, respectively.

During the event, $CN_{2.5}$ and $CN_{10}$ increased to 5669 and 5097 cm$^{-3}$, respectively. Furthermore, the

median $N_{NUC}$, $N_{AIT}$, and $N_{ACC}$ values were 508, 376, and 66 cm$^{-3}$, respectively. Elevated CCN concentrations at 0.2 and 0.4 % supersaturations were not observed, whereas CCN concentrations at 0.6, 0.8, and 1.0 % supersaturations slightly increased during the event. For instance, CCN concentration at 0.8 % supersaturation was 517 cm$^{-3}$ at 20:00 on January 13, then increased to 688 cm$^{-3}$, until 23:00 on January 13. The CCN concentration at 0.6, 0.8, and 1.0% supersaturations increased by 11%, 33%, and 58%, respectively.

**3.2.3. Multiple NPF event**

An intensive NPF event occurred from November 16 to November 17, 2018 (Figure 8). Air temperature during the event ranged from -2.5 to -0.1 °C (median -1.3 °C). RH ranged from 70–95% (median 79%), slightly lower than that for the marine and sea-ice NPF events described above. During the NPF event observed from 20:00 on November 16 to 02:00 on November 17, solar radiation decreased from 30 to 0 W m$^{-2}$. This suggests that the actual formation and growth occurred during daylight hours upwind from measurement location, but very slow growth continued over the Antarctic Peninsula allowing the detection of observed grown mode at ~ 7 nm after the sunset. Wind speed ranged from 4.3– 9.5 m s$^{-1}$ with a constant direction from the southwest (median 239 °). Air mass back trajectories showed multiple origins before reaching the station, passing over ocean (25.7 h, 53% of residence time), sea-ice (12.4 h, 26% of residence time), and land (10.0 h, 21% of residence time) (Figure 5c). During the event, the median total DMSP and chlorophyll exposures in the sea surface were 6.0 nmol L$^{-1}$ and 0.2 mg m$^{-3}$, respectively.

At the start of the event (17:00 on November 16), $N_{NUC}$, $N_{AIT}$, and $N_{ACC}$ were 687, 83, and 13 cm$^{-3}$, respectively. The particle number concentration of the nucleation mode sharply increased to 1610 cm$^{-3}$ at the NPF time, and its peak concentration occurred 7 h after the start of the event (00:00 on November 17), indicating spatial extent of the formation region. The peak concentration of Aitken mode particles successively appeared 14 h after the start of the event (07:00 on November 17) and 22 h respectively for accumulation mode particles (15:00 on November 17). The values in the Aitken and accumulation mode

ranges were 448 and 92 cm$^{-3}$, respectively. We also observed a gradual increase in CCN concentration for

23 h. CCN concentration at 0.4 % supersaturation increased from 78 (17:00 on November 16) to 272 cm$^{-}$

$^{3}$ (15:00 on November 17). This NPF event may have been a source of CCN, which enhanced CCN

concentrations by 248%.

**3.3. Influence of air mass origin on the NPF event**

**3.3.1. Parameters related to NPF**

Our results show that NPF and its growth events had largely different features depending on air

mass origin (Figure S6). Although only 3 cases of multi-regional NPF events occurred during the pristine

and clean periods (not included in Figure 9), the most intense NPF event was observed with multi-regional

source region. Here, we compared $N_{NUC}$, FR GR, and CS, for the ocean and sea-ice air masses (Figure 9

a-d). The FR, GR, and CS values agreed well with those reported in previous studies at other Antarctic

sites (Järvinen et al., 2013; Kim et al., 2019; Kyrö et al., 2013; Weller et al., 2015), but significantly lower

than the values reported by the Quéléver et al. (2022), who showed the average FR and GR were 0.686

460 cm$^{-3}$ s$^{-1}$ and 4.2 nm h$^{-1}$, respectively. The median $N_{NUC}$ and FR values for the ocean air mass ($N_{NUC}$: 220

461 cm$^{-3}$ and FR: 1.2 $\times$ 10$^{-2}$ cm$^{-3}$ s$^{-1}$) were 1.6 and 3.0 times lower than those of sea-ice air mass ($N_{NUC}$: 343

462 cm$^{-3}$ and FR: 3.6 $\times$ 10$^{-2}$ cm$^{-3}$ s$^{-1}$), respectively. This implies that marine NPF events are frequent, but

weak in terms of $N_{NUC}$ and FR values. Unlike $N_{NUC}$ and FR, there were no marked differences between

the median values of GR and CS by air mass category. The median particle GR values for ocean, sea-ice,

and multiple air masses were 0.8, 0.7, and 0.9 nm h$^{-1}$, respectively. In comparison, Jokinen et al. (2018)

reported that GR values ranged from 0.3 to 1.3 nm h$^{-1}$ at Aboa, and Brean et al. (2021) showed GR of 0.4

to 0.6 nm h$^{-1}$ measured during the PI-ICE cruise. To examine the effects of oceanic biological activity on

NPF properties, we examined solar radiation intensity, chlorophyll exposure and DMSP exposure for the

two air mass (or source region) categories (Figure 9 e-g). There was no difference in the median value in

solar radiation, while the median values for chlorophyll exposure and DMSP exposure were higher in air

masses originating from the ocean than in air masses originating from the sea ice. The median chlorophyll exposure in ocean-influenced air masses (0.2 mg m$^{-3}$) was roughly twice that of the sea-ice-influenced air mass (0.1 mg m$^{-3}$). Total DMSP exposure for the ocean-influenced air mass was ~2.7 times that of the sea-ice air mass.

**3.3.2. Potential sources facilitating new particle formation**

The chlorophyll exposure and DMSP exposure during marine NPF events were higher than those during sea ice NPF events, suggesting a large chance to carry biologically-derived organic compounds from the open ocean areas to the observation site. DMSP, a metabolite of oceanic phytoplankton, is partly converted into gaseous DMS through enzymatic cleavage (Simó, 2001), which is the largest natural sulfur source in the atmosphere (Barnes et al., 2006). Hence, the photooxidation products of biogenic DMS in the Antarctic atmosphere (e.g., Sulfuric acid and Methane sulfonic acid) could be a major contributor to NPF and its growth when the air mass originates from the ocean. Jang at al. (2019) reported that NPF events were more frequent in air masses originating from the Bellingshausen Sea than the Weddell Sea during the biologically productive austral summer, and it is likely that the taxonomic composition of phytoplankton can be related to the formation of new particles in the Antarctic Ocean. Biogenic DMS was found to be a precursor of NPF in coastal Antarctica (Yu and Luo, 2010).

The air mass exposure to chlorophyll and DMSP for sea-ice NPF events were 1.8 and 2.7 times lower than those of marine NPF events. This could be explained by volatile iodine compounds released from ice-covered areas in Antarctica (Jokinen et al; 2018; Saiz-Lopez et al., 2007; Sipilä et al., 2016); however, iodine compounds were not measured during our study period. Previously, iodine compounds were found in large concentrations in and above the sea-ice of the Weddell Sea in Antarctica during spring and summer (Atkinson et al., 2012). Roscoe et al. (2015) also confirmed that iodine compounds may contribute to the secondary production of a significant number of particles measured at Halley and Neumayer on the Antarctic coast.

In our study, sea-ice NPF events occurred frequently in January (middle of austral summer) and

September (early austral spring) (Figure 4). We compared the JR, GR, and CS values for the sea-ice NPF cases observed between January and September (Figure S7) because of their notable differences in ice coverage. In Antarctica, the minimum ice coverage is observed in February and the maximum in September (Parkinson and Cavalieri, 2012). Our results showed that JR, GR, and CS values were much higher in January than in September, indicating different NPF processes. The January events occurred under low ice-coverage conditions, similar to previous studies from polar areas such as Svalbard (Dall'Osto et al., 2017) and Greenland (Dall'Osto et al., 2018). Both studies showed that NPF events are related to biogenic precursors released by open water and melting sea-ice regions, particularly during the summer. In contrast, the September events occurred under high ice-coverage conditions. The monthly median values of solar radiation showed that solar radiation intensity was very low from May to August and then started to increase from September (Table 1). During the September events, median solar radiation intensity was found to be 63 W m$^{-2}$. It is therefore possible that elevated sea-ice concentrations under sufficient solar radiation around Antarctica lead to an increase in the concentration of halogen species, resulting in the production of newly formed particles.

The most intensive NPF event was observed in multiple air masses, although the oceanic biological activity was lower than that in the oceanic air mass. This indicated that terrestrial sources from continental Antarctica, in addition to both DMS (mainly from the ocean) and iodine (mainly from sea-ice), may have contributed to NPF. First, previous studies have reported that precursors emitted from seabird colonies are linked to NPF (Schmale et al., 2013; Weber et al., 1998). The Chottaebawi area in the southwestern part of King George Island (around 2 km away from our observation site) is an important penguin colony in the Antarctic region (Lee et al., 2009), while the cape area near King Sejong Station is abundantly populated by flying seabirds such as skua. Given the proximity and abundance of seabird colonies at King Sejong Station, seabird colony emissions are the likely sources of precursor gases to NPF (e.g., ammonia and amine) (Quéléver et al., 2022). In fact, 2 NPF events (4 February 2018 for marine air mass origin and 18 February 2018 for multiple air mass origin) were observed when winds were seen to originate from the south sector where strong emission from the penguin colonies (southeast sector of 106–140º). Figure

S8 showed the contour plots of the size distributions and wind roses during those days. Although we did not directly measure the precursor gases such as ammonia and amine that can trigger the NPF, we can speculate that the fauna on the land or at the shore such as penguin and seabird colonies could not be excluded as the potential source of NPF events locally although highly productive and ice melting Weddell sea is coinciding with southeast direction too. Previous studies reported that precursor gases for NPF (e.g., ammonia) can originate from the decomposition of excreta from seabirds and penguins (Lachlan-Cope et al., 2020; Legrand et al., 1998; Liu et al., 2018; Schmale et al., 2013). More recently, Quéléver et al. (2022) proposed that nitrogen-containing species could be land-sourced (e.g., from a high penguin population during the summertime) or marine-sourced (e.g., from the biological activity of plankton in the ocean and melting sea ice). The ammonia from seabird-colony guano is a key factor contributing to bursts of newly formed particles, which are observed in the summertime Arctic (Croft et al., 2016). Second, biogenic emissions from vegetation in the Antarctic Peninsula, mainly composed of relatively small and sparse patches of lichens and mosses (Miranda et al., 2020), could be associated with NPF and growth. Kim et al. (2006) studied plant communities on the Barton Peninsula around King Sejong Station in the maritime Antarctic and concluded that ~47% of the investigated area was covered by vegetation although generally sparse. Finally, biogenic precursors from meltwater ponds in continental Antarctica have also been suggested (Kyrö et al., 2013) as a possible source of aerosol production (Weller et al., 2018). Overall, our data suggest that complex interconnected ecosystems across ocean, sea ice, and land can lead to an enhancement in Antarctic NPF.

### 3.3.3. NPF as a source of CCN

For a given SS of 0.4%, the median CCN number concentrations were rather similar 184, 144, and 178 $cm^{-3}$ for ocean, sea ice, and multiple air masses, respectively (Figure 9h). The values are in line with previous studies published from the Antarctic regions. Humphries et al. (2023) reported CCN concentrations nearby East Antarctic observations from Macquarie Island and Kennaook / Cape Grim as well as recent ship voyages of the RSV Aurora *Australis* and the RV *Investigator* in the region. The median CCN value at a SS of 0.5% was in the ranges of 88–145 $cm^{-3}$ at Macquarie Island, 57–158 $cm^{-3}$ at

Kennaook / Cape Grim, and 40–230 cm$^{-3}$ during the voyages (No voyage data exist for the winter months),

respectively. The PCAN project exhibited that a median particle number concentration larger than 3 nm

of 354 cm$^{-3}$ was observed from the voyage and median CCN at 0.55 % supersaturation were 167 cm$^{-3}$,

implying approximately half the particles measured as CN$_3$ could be activated as CCN (Simmons et al.,

2021). Recently, several ship-based measurements over the Southern Ocean found significantly increased

MSA concentrations in air masses originating close to the Antarctic coastline, alongside enhancements in

CCN concentration (Humphries et al., 2021). Of the 83 NPF events, CCN concentrations increased by 2–

268% (median 44%) following 1 to 36 hours (median 8 hours) after NPF events. The median increase in

CCN concentrations was 44 %, 34 %, and 107 % for ocean, sea ice, and multiple air masses, respectively.

NPF can be an important source of CCN in Antarctica, and the highest CCN enhancement was observed

when air masses passed through multiple regions, followed by ocean and sea-ice regions. Our results

provide the first direct evidence of CCN production resulting from an NPF event in the Antarctic

atmosphere, based on simultaneous measurements of particle number size distribution (e.g., diameter

ranges of 2.5–300 nm) and CCN number concentrations in real time throughout the year.

We also compared CCN activity and critical diameter for the three selected periods (Figure 9i and

j). The median values of CCN activity, i.e. the ratio of the number concentration of particles that activated

to become CCN at a given supersaturation to the total number concentration or particles larger than 10

566    nm (CN$_{10}$), were similar (about 5%) in three different air masses. The critical diameter ($D_c$), the diameter

at which the integration of aerosol size distribution from the largest particle diameter to the lowest one

matches the measured CCN concentration, was determined using the measured aerosol size distribution,

CN$_{10}$, and CCN concentrations (Furutani et al., 2018). The median $D_c$ value at 0.4% supersaturation was

estimated to be 41 nm, 32 nm, and 37 nm for ocean, sea ice, and multiple air masses, respectively. These

results agreed well with those reported in previous studies that determined $D_c$ at the Finnish Antarctic

Research Station, Aboa (Kyrö et al., 2013), a clean subarctic background site (Komppula et al., 2005),

and over remote Southern Ocean around Antarctica (Fossum et al., 2018; Fossum et al., 2020). For

instance, Kyrö et al. (2013) found the smallest $D_c$ at 48 nm. The median $D_c$, as suggested by Komppula

et al. (2005), varied from 50–128 nm (average 80 nm). The $D_c$ value for maritime polar and marine modified continental Antarctic air masses were 71 and 59 nm, respectively (Fossum et al., 2020).

A time series of daily mean CCN concentrations at five different supersaturation ratio of 0.2, 0.4, 0.6, 0.8, and 1 was illustrated in Figure S10. To understand the contribution of growing particles on the CCN concentrations during NPF event, we determined the increase in CCN concentration during growth periods (i.e., growth to smaller than 40 nm particles and growth to larger than 40 nm particles) compared to baseline values under different supersaturation conditions (Figure 10), according to the method suggested by Chang et al. (2022). When particle growth was smaller than to 40 nm (growth ≤ 40 nm), the mean CCN concentrations increased by 59–178 cm$^{-3}$ for a SS of 0.2 %–1.0 %, representing a 172–217 % increase compared to the values during baseline conditions. When particle growth was larger than to 40 nm (growth > 40 nm), the mean CCN concentrations increased by 57–227 cm$^{-3}$ for a SS of 0.2 %–1.0 %, representing a 170–249 % increase compared to baseline values. Our results indicate that particles formed from NPF events can lead to the significantly enhanced CCN concentration in Antarctic Peninsula, and this effect is more pronounced if we consider particle growth larger than 40 nm, consistent with ship-based observations (Chang et al., 2022) and aircraft-based observation (Willis et al., 2016) in the Canadian Arctic during summer.

## 4. Conclusions

We measured the number size distribution of 2.5–300 nm particles and CCN number concentrations at King Sejong Station in the Antarctic Peninsula continuously from January 1 to December 31, 2018. The annual median values of $N_{NUC}$, $N_{AIT}$, and $N_{ACC}$ were 46.8 cm$^{-3}$, 53.5 cm$^{-3}$, and 21.7 cm$^{-3}$, respectively. Overall, clear annual and seasonal patterns of particle number concentrations in all size classes were observed (high concentrations in summer and low concentrations in winter). Furthermore, the monthly $CN_{10}$ value was positively correlated with the monthly $N_{NUC}$, implying that summer maximum particle concentrations could be largely influenced by newly formed particles in the Antarctic atmosphere. Among meteorological parameters, the elevated $N_{NUC}$ values (i.e., indicators of NPF events) were more likely to

be accompanied by high solar radiation, high temperature, and low RH, regardless of wind speed.

NPF events were identified based on size distribution data measured using two SMPSs. During the pristine and clean periods, 97 events (26% of observation days) with elevated $N_{NUC}$ were observed. NPF events occurred more frequently in summer than in any other season. Based on air mass back-trajectory analysis, we distinguished three different types of NPF events: marine (80 cases), sea ice (12 cases), and multiple (3 cases). Marine NPF events were frequent and weak ($N_{NUC}$, 220 cm$^{-3}$; FR = 1.2 $\times$ 10$^{-2}$ cm$^{-3}$ s$^{-1}$; GR = 0.8 nm hr$^{-1}$) and occurred when the air mass exposure to oceanic phytoplankton was high (chlorophyll, 0.2 mg m$^{-3}$; DMSP, 18 nmol L$^{-1}$). The photooxidation of biogenic DMS in the Antarctic atmosphere could be a major contributor to marine NPF events. In contrast, sea-ice NPF events ($N_{NUC}$, 343 cm$^{-3}$; FR, 3.6 $\times$ 10$^{-2}$ cm$^{-3}$ s$^{-1}$; GR, 0.7 nm h$^{-1}$) were observed when the air mass exposure to oceanic phytoplankton was relatively low (chlorophyll, 0.1 mg m$^{-3}$; DMSP, 7 nmol L$^{-1}$), which may be due to volatile iodine compounds released from ice-covered areas. Strong NPF events ($N_{NUC}$, 516 cm$^{-3}$; FR, 3.2 $\times$ 10$^{-2}$ cm$^{-3}$ s$^{-1}$; GR, 0.9 nm hr$^{-1}$) were associated with multiple air masses, indicating complex interconnected ecosystems leading to an enhancement in Antarctic NPF.

To investigate the connection between newly formed particles and CCN production, we compared CCN properties for the three air mass categories. The median CCN number concentrations at a given SS of 0.4% were 184, 144, and 178 cm$^{-3}$ for ocean, sea ice, and multiple air masses, respectively. Of the 83 events, an increase in CCN concentrations after the NPF events was detected, ranging from 2–268 % (median 44 %). The median increase in CCN concentrations was 44 %, 34 %, and 107 % for ocean, sea ice, and multiple air masses, respectively. NPF events led to increased CCN concentrations at King Sejong Station. The median value of $D_c$ at a supersaturation of 0.4% was estimated to be 41 nm, 32 nm, and 37 nm for ocean, sea ice, and multiple air masses, respectively. This study is the first to report CCN production resulting from Antarctic NPF events in the Antarctic Peninsula. However, further detailed measurements of the chemical properties of aerosol particles and precursor gases (e.g., ammonia) during NPF events are required to better understand the contribution of these compounds to the formation and growth of aerosol particles and to explore their impacts on CCN formation in the remote Antarctic

environment.

**Data availability**

The data analyzed in this publication will be readily provided upon request to the corresponding author

(yjyoon@kopri.re.kr).

**Author contributions**

JP and YJY designed the study. JP, HK, YG, EJ, K-TP, SP, and YJY analyzed data. JP wrote the

manuscript. CHJ, DC, and CO'D all commented on and discussed the manuscript.

**Competing interests**

The authors declare that they have no conflict of interest.

**Acknowledgments**

We would like to thank the many technicians and scientists of the overwintering crews. This work was

supported by the KOPRI project (PE23030).

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

Table 1. Monthly median for total particle number concentration > 10 nm ($CN_{10}$), particle number concentrations of the nucleation mode ($N_{NUC}$), Aitken mode ($N_{AIT}$), accumulation mode ($N_{ACC}$), CCN number concentration at supersaturation of 0.4% ($CCN_{0.4\%}$), and meteorological parameters such as solar radiation, temperature, RH, pressure, wind speed, and wind direction for 2018, after data filtering (BC < 50 ng m$^{-3}$ indicating pristine and clean conditions), measured at King Sejong Station in the Antarctic Peninsula from January 1 to December 31, 2018.

| | $CN_{10}$ (cm$^{-3}$) | $N_{NUC}$[a] (cm$^{-3}$) | $N_{AIT}$[a] (cm$^{-3}$) | $N_{ACC}$[a] (cm$^{-3}$) | $CCN_{0.4\%}$ (cm$^{-3}$) | Solar radiation (W m$^{-2}$) | Temp. (°C) | RH (%) | Pressure (hPa) | Wind Speed (m s$^{-1}$) | Wind direction (°) |
|---|---|---|---|---|---|---|---|---|---|---|---|
| January | 506.2 | 101.1 | 188.7 | 83.8 | 235.2 | 129.2 | 1.1 | 88.6 | 986.0 | 5.78 | 315.8 |
| February | 594.3 | 111.3 | 200.0 | 69.9 | 229.8 | 103.5 | 1.8 | 90.8 | 987.2 | 7.72 | 319.9 |
| March | 357.3 | 86.0 | 112.4 | 42.1 | 138.7 | 58.0 | 1.1 | 88.4 | 981.8 | 8.21 | 342.3 |
| April | 184.1 | 49.9 | 39.1 | 17.5 | 58.6 | 26.2 | -0.7 | 87.1 | 988.2 | 7.88 | 350.0 |
| May | 106.7 | 25.1 | 23.8 | 14.2 | 51.1 | 7.3 | -2.3 | 81.8 | 990.1 | 7.34 | 277.7 |
| June | 75.9 | 12.2 | 12.5 | 9.2 | 35.4 | 3.4 | -4.1 | 88.4 | 995.9 | 7.21 | 339.8 |
| July | 84.3 | 28.2 | 16.8 | 11.6 | 39.1 | 5.5 | -2.9 | 86.5 | 992.2 | 9.08 | 300.8 |
| August | 109.8 | 39.3 | 19.6 | 14.8 | 52.1 | 21.8 | -3.3 | 85.9 | 986.2 | 8.57 | 327.8 |
| September | 266.4 | 123.8 | 51.3 | 20.9 | 79.3 | 65.6 | -3.6 | 86.5 | 992.6 | 9.52 | 313.2 |
| October | 287.0 | 88.9 | 62.0 | 26.9 | 105.3 | 122.1 | -2.1 | 84.6 | 994.4 | 6.50 | 290.8 |
| November | 498.2 | 79.3 | 136.8 | 46.1 | 150.3 | 143.3 | -0.6 | 89.3 | 980.0 | 7.59 | 307.9 |
| December | 511.9 | 193.5 | 227.6 | 67.7 | 189.1 | 136.5 | 0.4 | 87.2 | 980.4 | 6.72 | 302.7 |

[a]$N_{NUC}$, $N_{AIT}$, and $N_{ACC}$ represent the particle number concentrations in the nucleation mode (2.5–25 nm), Aitken mode (25–100 nm), and accumulation mode (100–300 nm).

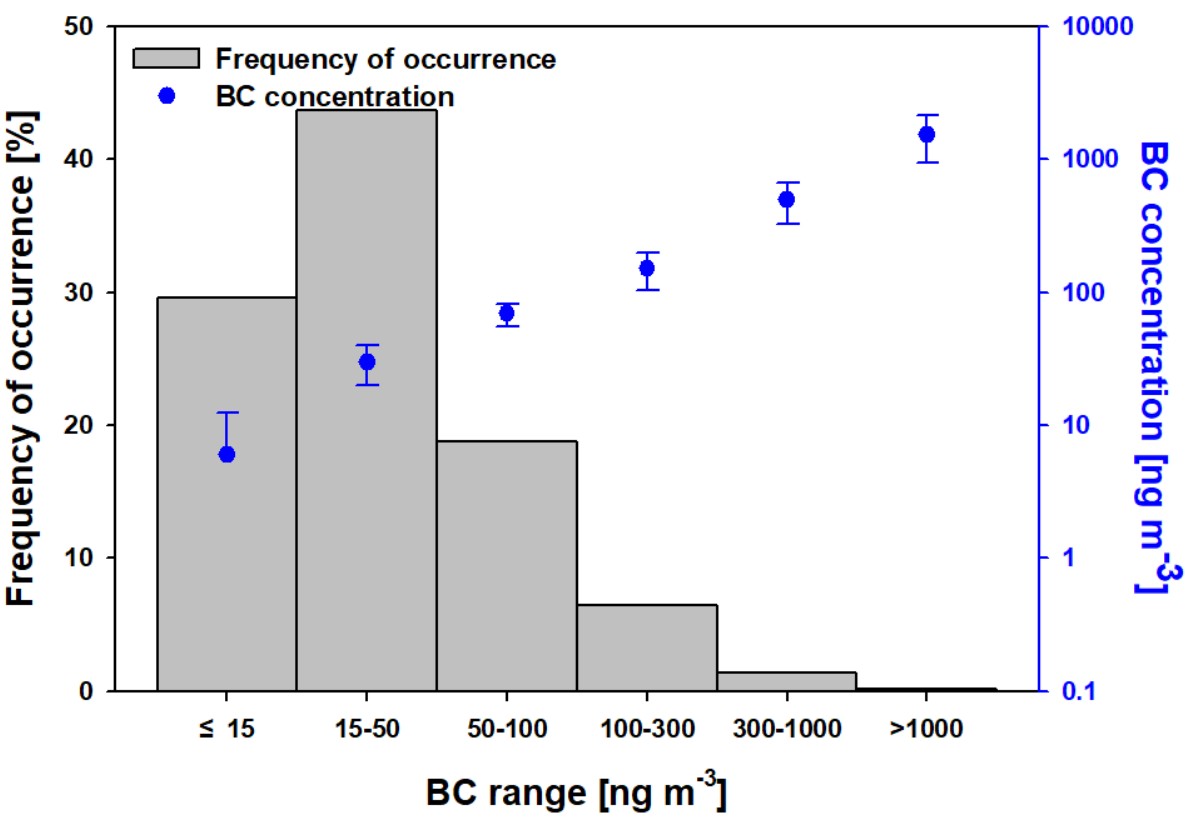

Figure 1. Frequency of occurrence of BC mass concentration for six types of Antarctic Peninsula air-pollution levels classified from four-year BC data.

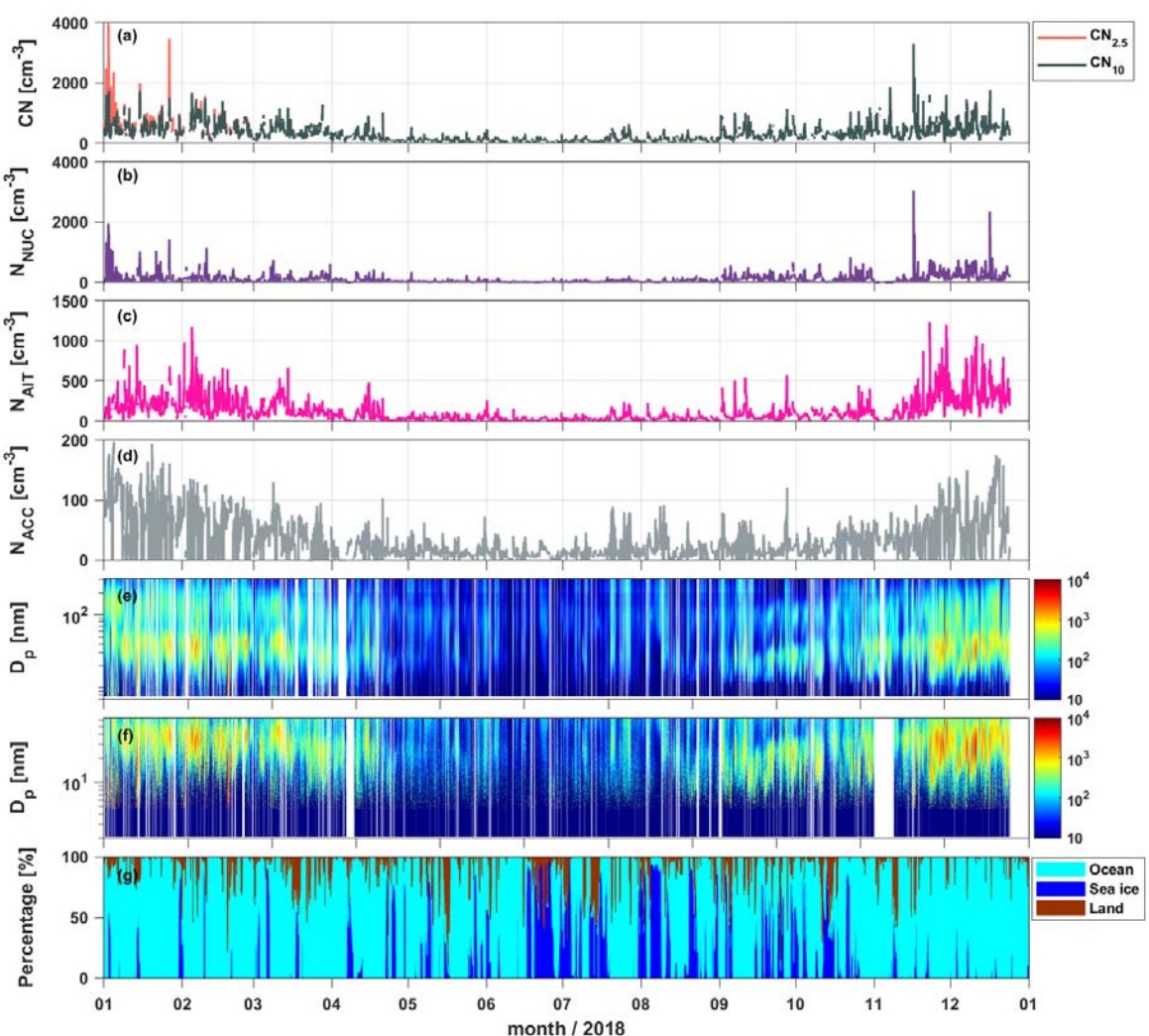

Figure 2. Time series of 1 h averages for (a) $CN_{2.5}$ and $CN_{10}$, (b) nucleation mode ($N_{NUC}$; 2.5–25 nm), (c) Aitken mode ($N_{AIT}$; 25–100 nm), and (d) accumulation mode ($N_{ACC}$; 100–300 nm); contour plots of the size distributions measured using (e) standard and (f) nano-SMPS; and (g) residence time of air masses passing over ocean, sea ice, and land. CN 2.5 data are only available from January to March due to the instrumental malfunctions.

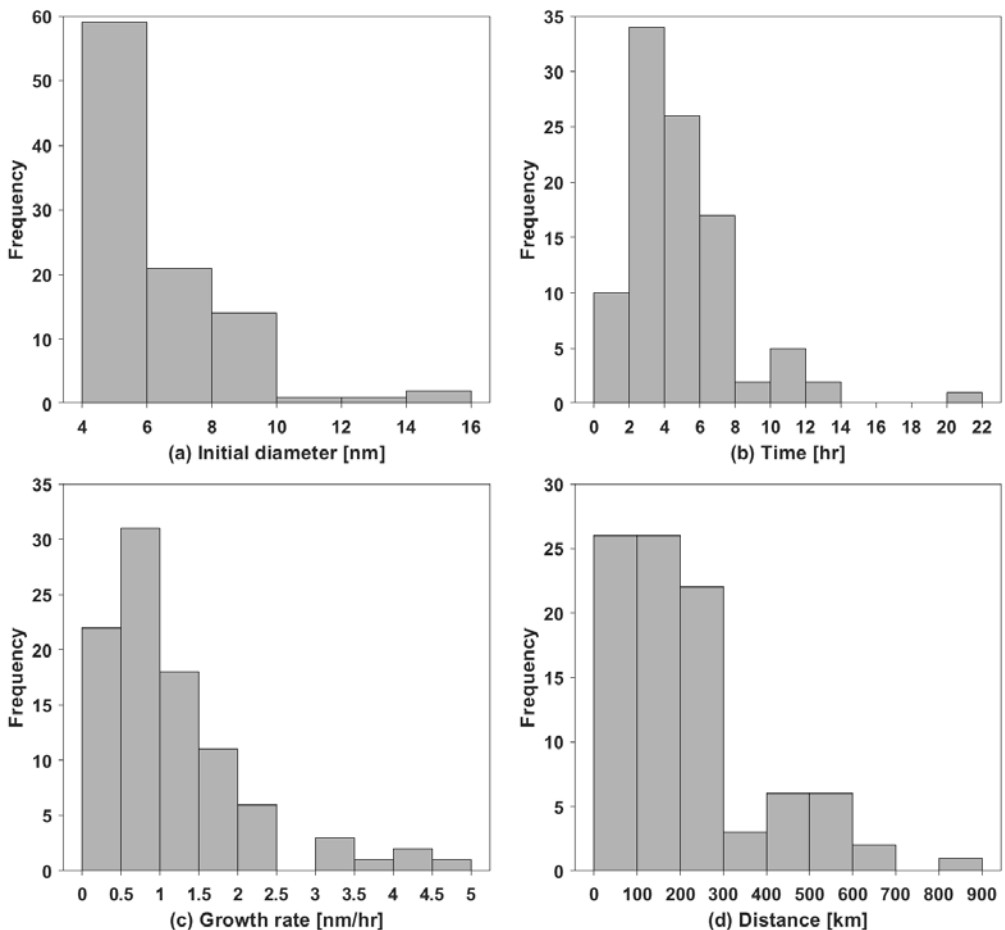

Figure 3. Frequency of (a) initial diameter of particles, (b) duration time, (c) growth rate, and (d) extension for the NPF event. Two NPF cases were excluded when the wind speed was higher than 10 ms$^{-1}$.

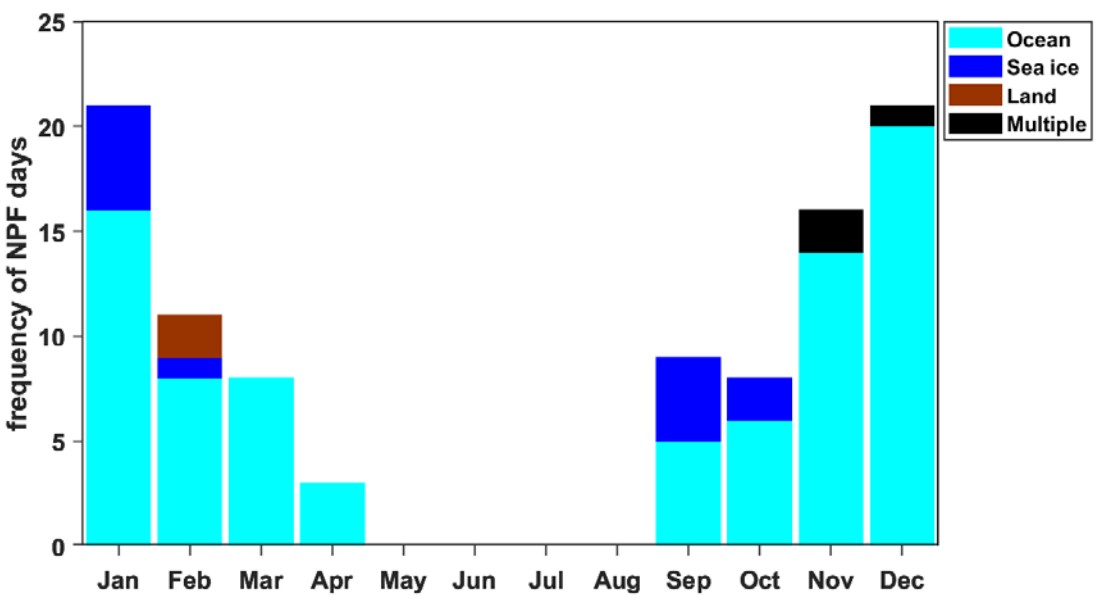

Figure 4. Seasonal variations in the number of NPF days by air mass origin.

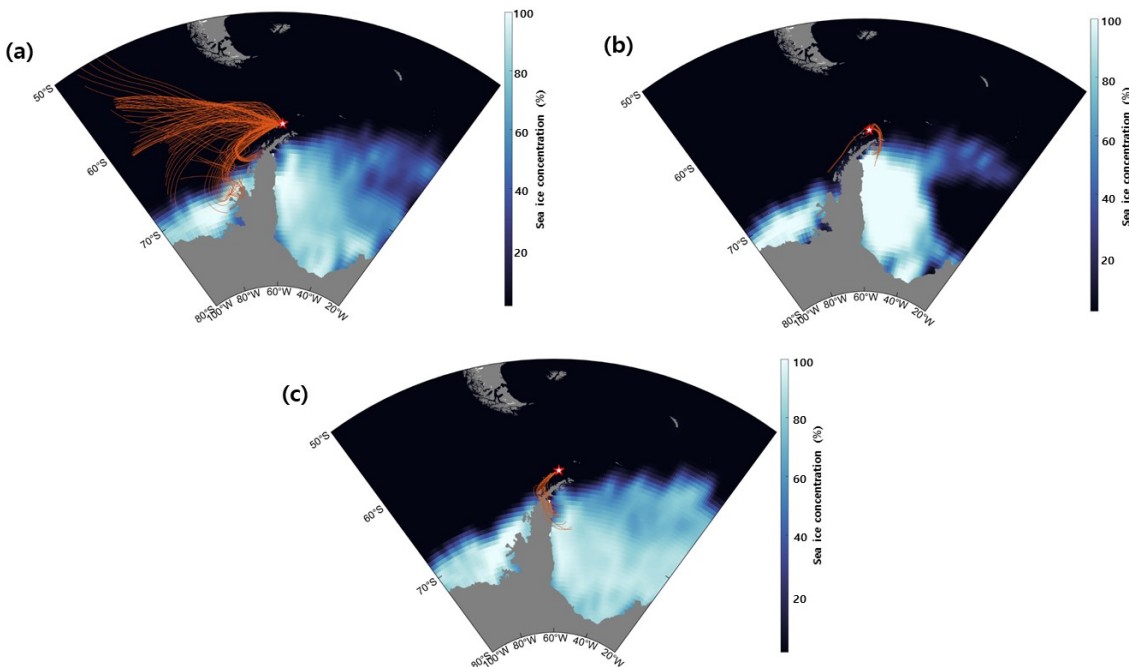

Figure 5. Air mass back trajectories for arrival at 50 m for the three case study NPF events: (a) marine, (b) sea ice, and (c) multiple.

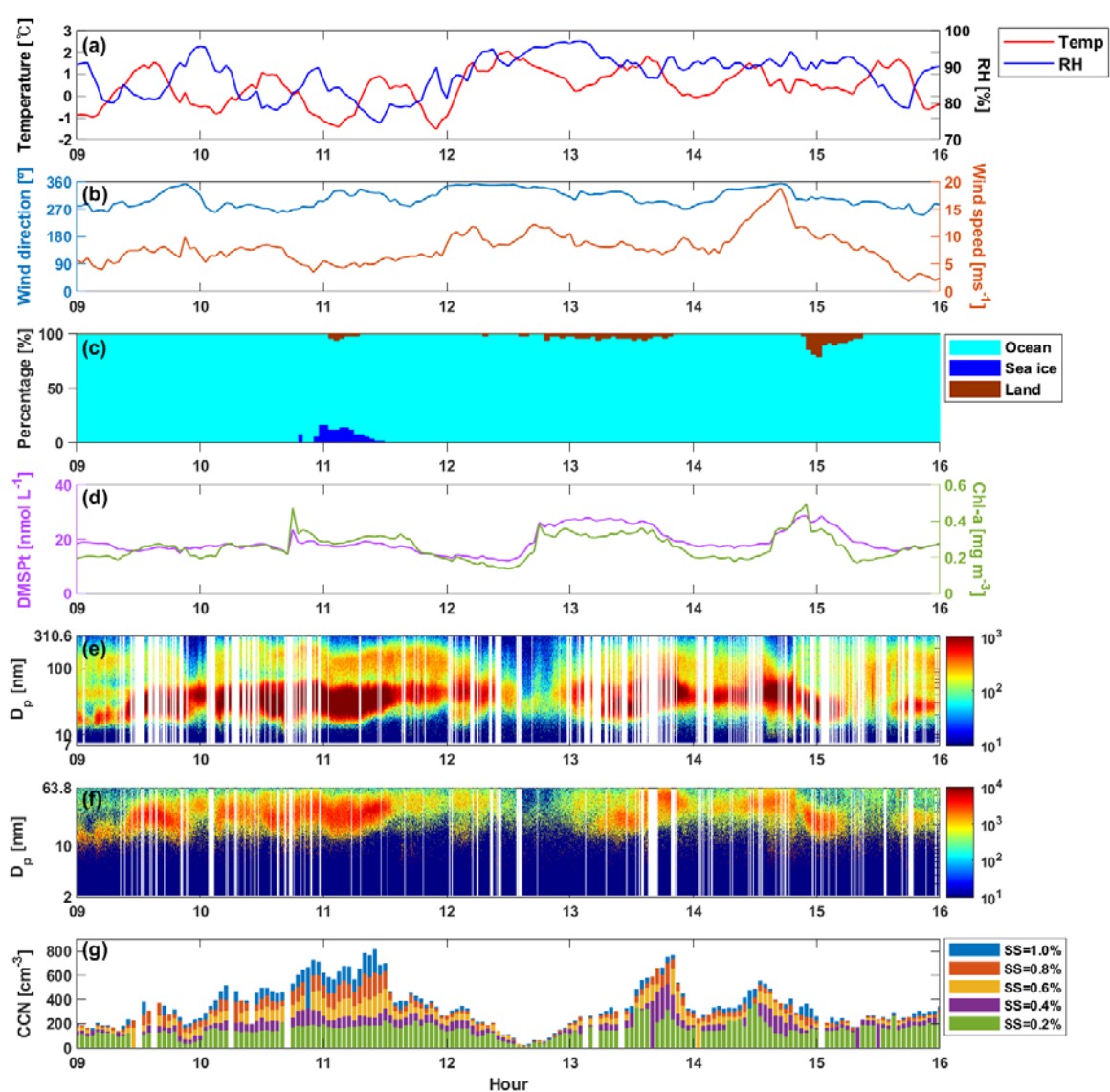

Figure 6. Marine NPF event observed from December 9–15, 2018. (a–b) meteorological variables, (c) the residence time of air masses that passed over the ocean, sea ice and land areas, (d) total DMSP and chlorophyll exposures, (e–f) number size distribution with the standard-SMPS and nano-SMPS, and (g) CCN number concentration. The x-axis represents local time.

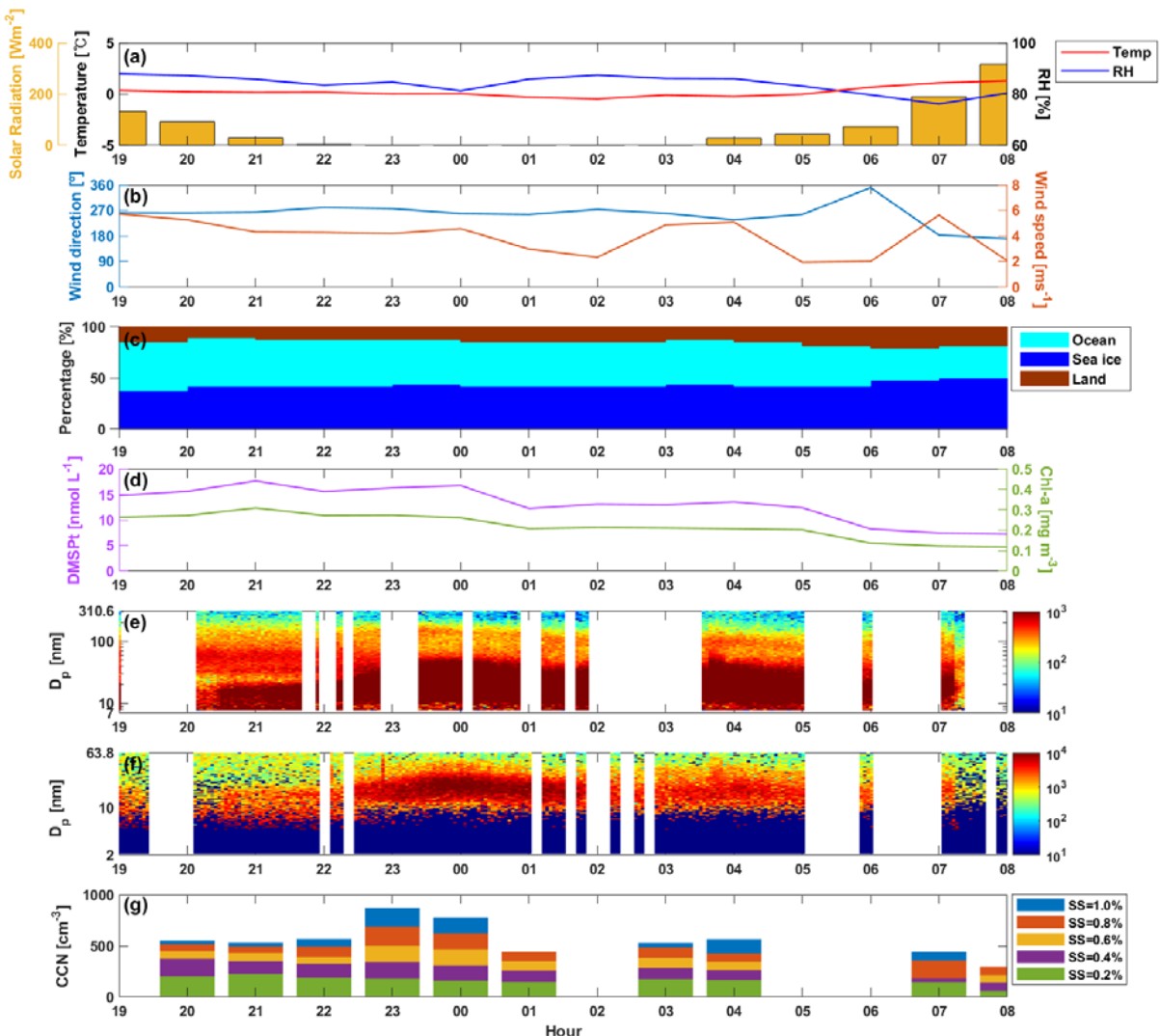

Figure 7. Sea ice NPF event observed from January 13–14, 2018. (a–b) meteorological variables, (c) the residence time of air masses that passed over the ocean, sea ice and land areas, (d) total DMSP and chlorophyll exposures, (e–f) number size distribution with the standard-SMPS and nano-SMPS, and (g) CCN number concentration. The x-axis represents local time.

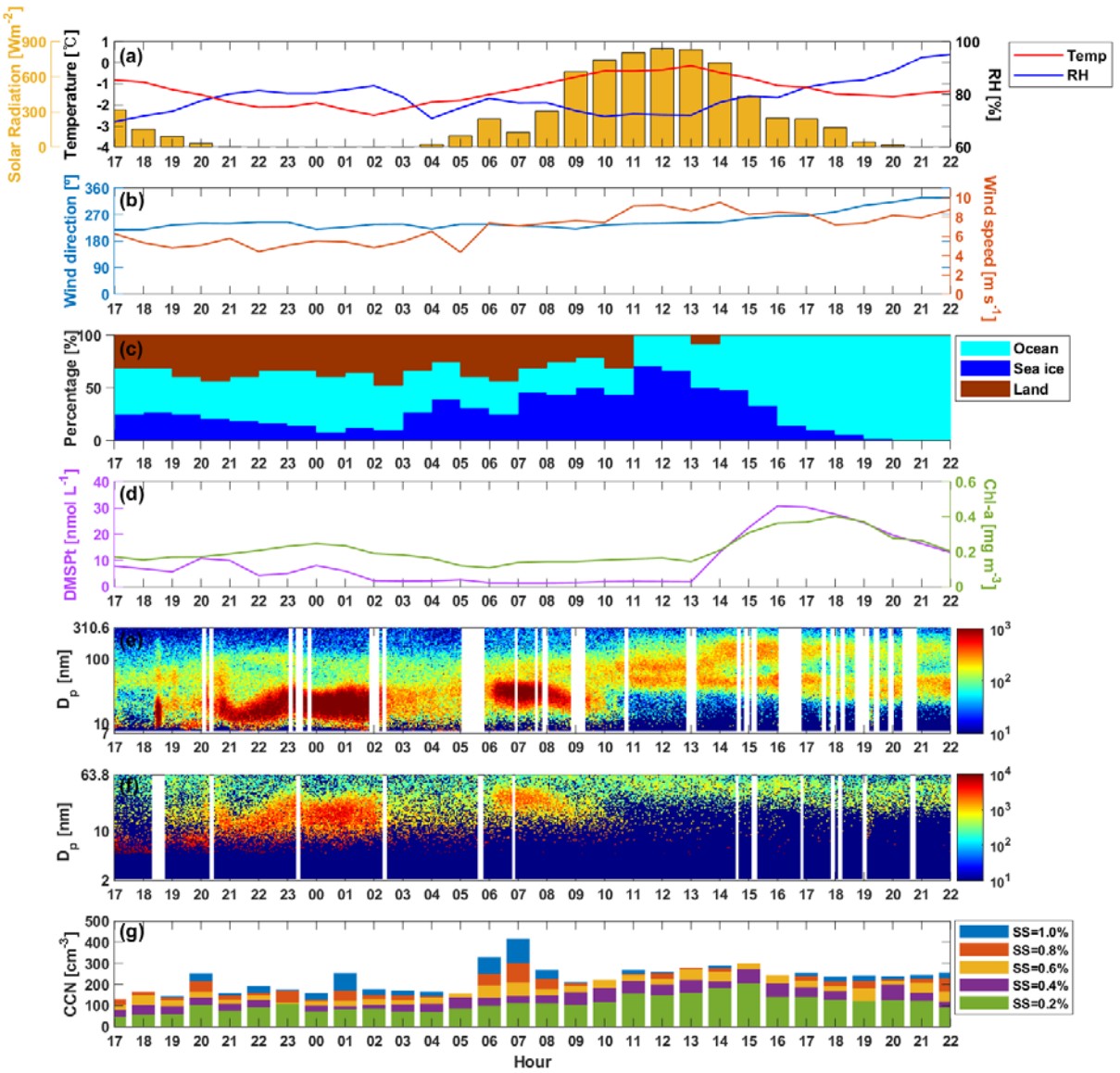

Figure 8. Multiple NPF event observed from November 16–17, 2018. (a–b) meteorological variables, (c) the residence time of air masses that passed over the ocean, sea ice and land areas, (d) total DMSP and chlorophyll exposures, (e–f) number size distribution with the standard-SMPS and nano-SMPS, and (g) CCN number concentration. The x-axis represents local time.

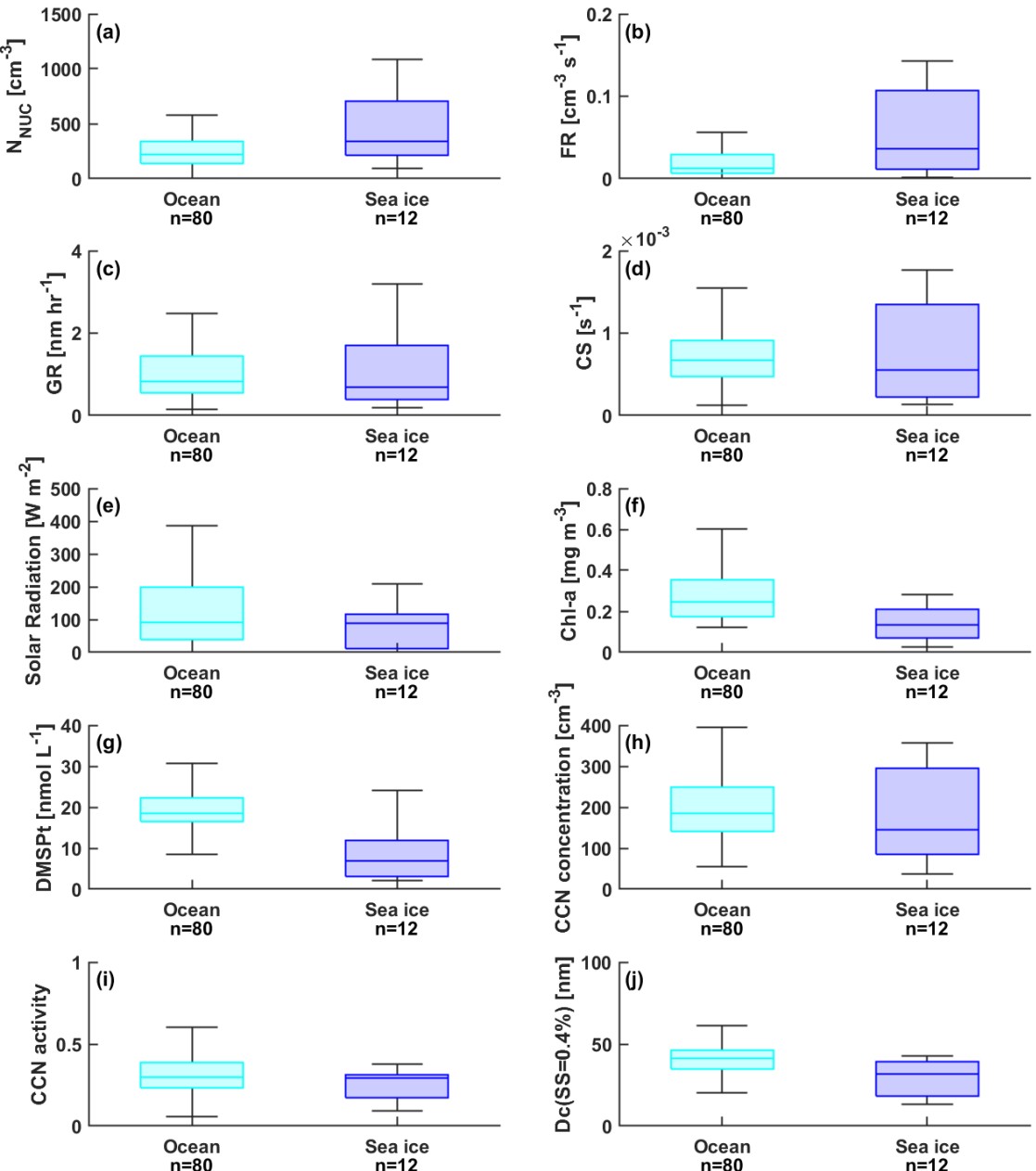

Figure 9. Box plots for (a) number concentration of nucleation-mode particles ($N_{NUC}$), (b) formation rate (FR), (c) growth rate (GR), (d) condensation sink (CS), (e) solar radiation, (f) chlorophyll exposure, (g) DMSP exposure, (h) CCN number concentration, (i) CCN activity, and (j) critical diameter ($D_c$) for ocean, sea ice, and multiple air masses. Upper/lower box limits and solid lines indicate the 75th/25th percentiles and median, respectively.

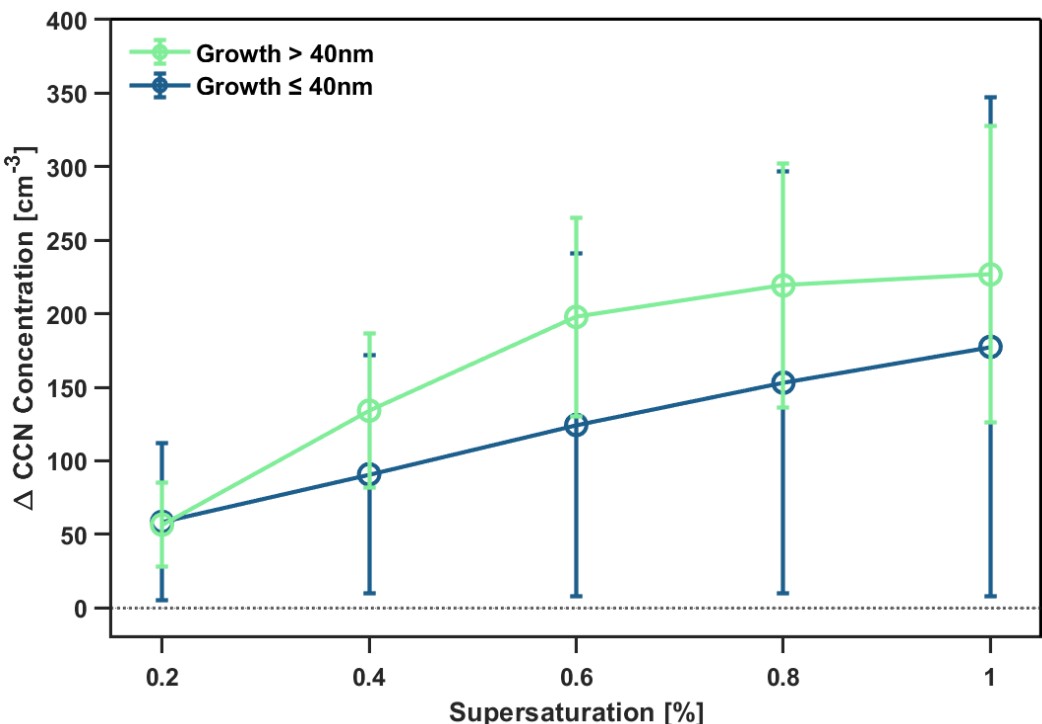

Figure 10. The increase in CCN concentration during growth to larger than 40 nm particles (green) and smaller than 40 nm (blue) times compared with background times at five different supersaturations.