# Peer review of "New particle formation leads to enhanced cloud condensation nuclei concentrations in Antarctic Peninsula"

_EGUsphere, 2023_

## Referee Comment (RC1)

**Review of "*New particle formation leads to enhanced cloud condensation nuclei concentrations at Antarctic Peninsula*" by Park et al.**

This study tackles the result of continuous size distribution and cloud condensation nuclei (CCN) measurements in the King Sejong research station located North of the Antarctic Peninsula for the whole year 2018. The work presents a consistent and continuous set of data for the physical characterization of aerosol particles which enabled to assess the significant occurrence of NPF, principally during the summertime. Authors addressed the new particle formation events observed (97 in total), as well as the source point – specifically looked out among marine, sea-ice, a multiple-origin study cases -, and discussed possible chemical drivers of the observed new particle formation events. Finally, the study focused on the CCN result in relation to the observed NPF. The paper is generally well written, and I suggest the publications in ACP after the revisions/clarifications on the following points:

**General comments:**

The study undeniably complements the knowledge on NPF and its occurrence in the remote Antarctic field. In the context of polar region, NPF statistic based on seasonality would be an asset while showing occurrence during summer / winter and the transitions period of the melting and refreezing of the ocean. (Suggestions: Examples of 'typical' event, with size distribution surface plot would introduce nicely the discussed topic).

The source attribution was thoroughly investigated and well described in the study case subsections. Without direct measurement of precursor gases at site, potential source for NPF can only be discussed rather than undeniably explained, which clearly expressed in the manuscript. However, one big argument is the local fauna, and whose emission surely influence the observation, and this possibly independently of the air trajectory due to its close vicinity. Discussion could be developed in that regard (Suggestion: the use wind data could then be relevant).

The contribution of NPF to CCN would find benefits in a strong(er) linkage rather than with observations of increased CCN after occurrence of NPF solely, possibly using the hygroscopicity factor in relation to growing particles. Alternatively, the authors could rise conscience in the missing link between the two datasets and the need for dedicated studies/measurement in the future.

**Specific Comments (RC):**

RC$_1$ (**Abstract**) Since the study present a year-long dataset of size distribution, one could expect general information on the seasonality and frequency of NPF observed throughout the year. This would bring context and significance to the abstract.

RC$_2$ (**§1, L.53-L.55**) Have the authors considered possible anthropogenic activity as well as the newly emerging land vegetation as a possible source of NPF to be mentioned in introduction as well.

RC$_3$ (**§1, L.57-L.58**) I suggest adding Sipilä et al. (2016, https://doi.org/10.1038/nature19314) as a key reference for showing the role of Iodic acid in NPF in Marine & Polar environments.

RC$_4$ **(§1, L.71-L.75)** Let me bring to your attention the publication of Quéléver et al. (2022, https://doi.org/10.5194/acp-22-8417-2022) reporting new particle formation at the Marambio station during the early 2018 (incl. measurement period of the present study), also reporting particle characterization based on size distribution. This reference could also bring context in the discussion presented later in **(§3.3.1 and §3.3.2)** regarding the relationship between NPF occurrence and meteorological parameters and potential (chemical) source of NPF.

RC$_5$ **(§2.1, L.107-L.108)** Please clarify for each CPCs if this corresponds to measurement incl. ultrafine particle (corresponding data CN$_{2.5}$) or particles larger than 10 nm (corresponding data CN$_{10}$), if mentioned already in the method, there is then no need to specify it later (e.g., **L.233 – L.234** or in the caption of Fig.2).

RC$_6$ **(§2.2, L.155-L.159 & L.165-L.168)** Could you explain the reason for the BC comparison between North Atlantic Ocean and Southern Ocean (where northern hemisphere is usually more influence by anthropogenic factors)? The environment being significantly different, I would suggest to clarify the context or to revise the relevance of this additional information to the manuscript.

RC$_7$ **(§3.1)** I would suggest to re-assess the structure of the subsections within 3.1 in order the easily follow the story line of the analysis by, for example, adding a subsection for the meteorological parameters influencing the aerosol particles and their formation (L. 248).

RC$_8$ **(§3.1.1, Table 1 & L.230)** Please reformulate the caption for Table 1: e.g., "*Monthly median for total particle number concentration >10 nm (CN10) ....*". I also suggest to add, in the caption, that the data are filter for pristine / clean conditions only (i.e., data when BC <50 ng.m-3). Finally, review the sentence **L.230** accordingly as **Table 1** does not show the time series for one-hour average but it recaps monthly median values for the year 2018.

RC$_9$ **(§3.1.1, Figure 2)** CN$_{2.5}$ is visible only from Jan. to Mar., it would be worth to comment on that in the text as well. Also consider to use color set that are color blindless-friendly (e.g., other than green and red in the same plot). **(L.244-L.247)** The correlation analysis CN$_{10}$ vs N$_{NUC}$ could be more relevant with a bigger data set rather than with monthly averaged values, why not using the one-hour data?

RC$_{10}$ **(§3.1.1, L.270-L.272)** Although the authors focus first on specific meteorological parameters influencing NPF, I suggest to strongly insist on the combination high PAR, high temperatures, low RH altogether rather than finding explanation with one parameter alone. Furthermore, high windspeed would enhance mixing of the emitted compounds and accelerate transport (possibly from further away). Please add reference for enhanced VOC in high windspeed conditions.

RC$_{11}$ **(§3.1.2, L.276 -)** The section 3.1.2 depicts the statistics on the observed NPF events. I would suggest to first mentions the numbers of event observed with a brief description of representative event types (if such grouping can be done), e.g., burst events, nucleation transported + local growth, etc.… incl. example with size distribution surface plot.

RC$_{12}$ **(§3.1.3, L.301 -)** The start of the section reintroduces the NPF / nucleation presented in the earlier section, I would recommend restructuring the section 3.1. in order to follow a coherent path on the descriptions of NPF events, without reintroducing NPF observation on every subsection.

RC$_{13}$ (**§3.2**) For each case study, the manuscript shows the result of CCN data for one supersaturation ratio only, could the authors develop on the reason for using this data only rather than comparing with the information brought with the other super saturation ratio.

RCx (**§3.3.2, L.426 -**) Here as well, I suggest adding Sipilä et al. (2016, https://doi.org/10.1038/nature19314) as a key reference for showing the role of Iodic acid in NPF in Marine & Polar environments.

RC$_{14}$ (**§3.3.2, L.452 -**) The presence of Penguins close (~2 km) to the measurement location is a determinant parameter impacting the frequency of NPF. Earlier in the section, the authors discussed extensively on the air mass origin, however this now bring a strong local source for chemicals bases such as ammonia that can trigger NPF. This, however, could only be validated by direct measurement of precursor gases. Hence, I highly recommend discuss this further, as well as the need for gas phase measurement. As such, a close emission point will likely interact with the result interpretation incl. from most / all the air mass origin. Furthermore, if pinguins / birds are only present in the surrounding seasonally, I would further suggest a comparison on the frequency / intensity of the NPF observed between e.g., breading season and start of the spring.

RC$_{15}$ (**§3.3.3, L.469 -**) Could the authors bring clarity on the connection between the size distribution data and the CCN data. It is a critical point of this study, as NPF and CCN do not occur at the same time, it would be interesting to account for parameters associated with CCN formation in connection to NPF (particle number/diameter, sinks, survival probability …) Have the authors considered CCN transported from another source (i.e., primary particles)? Alternatively, the authors could present the result given with supersaturation ratio and discuss the link with hygroscopicity (CCN data) and growing particles (DMPS data). I then suggest referring to Chang et al. (2022, https://doi.org/10.5194/acp-22-8059-2022) to complement the analysis.

**Technical comments (rmc):**

TC$_1$ (**Title**) Suggestion to revise the title by replacing the preposition "*at*" by "*in **the** Antarctic Peninsula*".

TC$_2$ (**Abstract, L.24-L.27**) Suggestion to reformulate as there is no direct measurement of DMS/DMSP. → E.g. "*Our estimation of DMPS concentration from satellite chlorophyl data suggest that product of biogenic precursor could be a component of marine NPF, ...*".

TC$_3$ (**§2.1, L.129-L.130**) CCNC supersaturations either 20%, 40%, 60%, 80% 100% or supersaturation ratio of 0.2, 0.4, 0.6, 0.8, 1.

TC$_4$ (**§3.1.3**) "*size spectra*" → "*size distribution spectra*"

TC$_5$ (**§3.1.3, L.390**) "evets"→ "events"

TC$_6$ (**§3.3.2, L.413 & L.418**) Please consider the product of oxidation of DMS are those responsible for the NPF. Condensable vapors: Sulfuric acid and Methane sulfonic acid.

---

## Author Comment (AC1)

**Reviewer #1: Review of "New particle formation leads to enhanced cloud condensation nuclei concentrations at Antarctic Peninsula" by Park et al.**

This study tackles the result of continuous size distribution and cloud condensation nuclei (CCN) measurements in the King Sejong research station located North of the Antarctic Peninsula for the whole year 2018. The work presents a consistent and continuous set of data for the physical characterization of aerosol particles which enabled to assess the significant occurrence of NPF, principally during the summertime. Authors addressed the new particle formation events observed (97 in total), as well as the source point – specifically looked out among marine, sea-ice, a multiple-origin study cases -, and discussed possible chemical drivers of the observed new particle formation events. Finally, the study focused on the CCN result in relation to the observed NPF. The paper is generally well written, and I suggest the publications in ACP after the revisions/clarifications on the following points:

We sincerely appreciate all valuable comments and suggestions, which helped us to greatly improve the quality of the manuscript. We corrected the manuscript point by point accordingly.

**General comments:**

The study undeniably complements the knowledge on NPF and its occurrence in the remote Antarctic field. In the context of polar region, NPF statistic based on seasonality would be an asset while showing occurrence during summer / winter and the transitions period of the melting and refreezing of the ocean. (Suggestions: Examples of 'typical' event, with size distribution surface plot would introduce nicely the discussed topic).

Response: Typical NPF events (e.g., burst event, nucleation with growth) were shown in Figure S3 as suggested by reviewer (Please, see reviewer's comment in RC11). To clarify the seasonality of NPF event, we included the average number size distribution of particles observed during summer (December to February), winter (June to August), and the transition periods of the melting of the ocean (September to November) and refreezing of the ocean (March to May) (Figure S4).

Page 13, Line 332: "In order to investigate the seasonal characteristics of NPF event, we compared mean size distributions of aerosol particles for summer, spring (transition period of the melting ocean), and autumn (transition period of refreezing of the ocean) (Figure S4). Trimodal distributions were presented in all seasons excepting winter when nucleation mode or particle formation was not observed. For instance, a trimodal distribution was seen at 7 nm, 30 nm, 122 nm during summer months. The number concentration of nucleation and Aitken modes were higher than the accumulation modes, indicating that NPF event regulates the aerosol processes in Antarctic peninsula. The largest mode at 126 nm or 103 nm may be due to a combination of primary (produced by bubble-bursting process) and secondary (produced by gas-to-particle conversion process) aerosol components. Results are broadly in line with previous results published from the Arctic and Antarctic regions. A ship-borne field campaign over Arctic Ocean found a trimodal distribution at 18 nm, 53 nm and 150 nm for open-ocean marine Arctic NPF event (Park et al., 2020). Lachlan-Cop et al. (2020) presented k-mean cluster analysis of particle size distribution measured at Halley, Antarctica, showing a nucleation peak at 15 nm for "nucleation" ultrafine category and a nucleation peak at 27 nm for "bursting" ultrafine category."

[Figure]

Figure S3. Mean size distributions of aerosol particles for summer (December–February) and transition periods of the melting of the ocean (September–November) and refreezing of ocean (March–May), when NPF event occurred. The average size distribution of aerosol particles for wintertime (June–August) was not displayed because we did not detect NPF events during the season.

The source attribution was thoroughly investigated and well described in the study case subsections. Without direct measurement of precursor gases at site, potential source for NPF can only be discussed rather than undeniably explained, which clearly expressed in the manuscript. However, one big argument is the local fauna, and whose emission surely influence the observation, and this possibly independently of the air trajectory due to its close vicinity. Discussion could be developed in that regard (Suggestion: the use wind data could then be relevant).

Response: Thanks for raising this issue. We did not measure the precursor gases. Out of 97 observed NPF events, 2 NPF events (4 February 2018 and 18 February 2018) were associated with local fauna. We presented the contour plots of the size distributions and wind roses when predominant wind possibly passing over a penguin colony (around 2 km away from our observation site) in Figure S8. We discussed the influence of local fauna such as penguin colonies on NPF event as given below.

Page 20, Line 513: "In fact, 2 NPF events (4 February 2018 for marine air mass origin and 18 February 2018 for multiple air mass origin) were observed when winds were seen to originate from the south sector where strong emission from the penguin colonies (southeast sector of 106–140°). Figure S8 showed the contour plots of the size distributions and wind roses during those days. Although we did not directly measure the precursor gases such as ammonia and

amine that can trigger the NPF, we can speculate that the fauna on the land or at the shore such as penguin and seabird colonies could not be excluded as the potential source of NPF events locally although highly productive and ice melting Weddell sea is coinciding with southeast direction too. Previous studies reported that precursor gases for NPF (e.g., ammonia) can originate from the decomposition of excreta from seabirds and penguins (Lachlan-Cope et al., 2020; Legrand et al., 1998; Liu et al., 2018; Schmale et al., 2013). More recently, Quéléver et al. (2022) proposed that nitrogen-containing species could be land-sourced (e.g., from a high penguin population during the summertime) or marine-sourced (e.g., from the biological activity of plankton in the ocean and melting sea ice). The ammonia from seabird-colony guano is a key factor contributing to bursts of newly formed particles, which are observed in the summertime Arctic (Croft et al., 2016)."

[Figure]

Figure S8. (a) Contour plots of the size distributions and (b) wind rose on 4 February 2018 and (c) contour plots of the size distributions and (d) wind rose on 18 February 2018. The southeast direction (106–140º) is designated as a sector where strong emission from the penguin colonies may originate. The x-axis represents local time.

The contribution of NPF to CCN would find benefits in a strong(er) linkage rather than with observations of increased CCN after occurrence of NPF solely, possibly using the hygroscopicity factor in relation to growing particles. Alternatively, the authors could rise conscience in the missing link between the two datasets and the need for dedicated studies/measurement in the future.

Response: The reviewer made a good point. To clarify the connection between growing particles and CCN, we determined the increase in the mean CCN concentration at different SS during growth times compared with background times based on the method by Chang et al. (2022). The sentence and figure were included as follows.

Page 22, Line 574: "To understand the contribution of growing particles on the CCN

concentrations during NPF event, we determined the increase in CCN concentration during growth periods (i.e., growth to smaller than 40 nm particles and growth to larger than 40 nm particles) compared to baseline values (black) under different supersaturation conditions (Figure 10), according to the method suggested by Chang et al. (2022). When particle growth was smaller than to 40 nm (growth ≤ 40 nm), the mean CCN concentrations increased by 59–178 cm$^{-3}$ for a SS of 0.2 %–1.0 %, representing a 172.3–216.7 % increase compared to the values during baseline conditions. When particle growth was larger than to 40 nm (growth > 40 nm), the mean CCN concentrations increased by 57–227 cm$^{-3}$ for a SS of 0.2 %–1.0 %, representing a 169.9–249.1 % increase compared to baseline values. Our results indicate that particles formed from NPF events can lead to the significantly enhanced CCN concentration in Antarctic Peninsula, and this effect is more pronounced if we consider particle growth larger than 40 nm, consistent with ship-based observations (Chang et al., 2022) and aircraft-based observation (Willis et al., 2016) in the Canadian Arctic during summer."

[Figure]

Figure 10. The increase in CCN concentration during growth to larger than 40 nm particles (green) and smaller than 40 nm (blue) times compared with background times at five different supersaturations.

**Specific Comments (RC):**

RC1 (**Abstract**) Since the study present a year-long dataset of size distribution, one could expect general information on the seasonality and frequency of NPF observed throughout the year. This would bring context and significance to the abstract.

Response: As suggested by reviewer, we added the information on the seasonality and frequency of NPF to the abstract.

Page 1, Line 20: "Clear annual and seasonal patterns of NPF were observed: high concentration and frequency of nucleation-mode particles in summer (December–February: 53 NPF cases)

and undetected nucleation-mode particles in winter (June–August: no NPF cases)."

RC2 (**§1, L.53-L.55**) Have the authors considered possible anthropogenic activity as well as the newly emerging land vegetation as a possible source of NPF to be mentioned in introduction as well.

Response: We mentioned additional possible sources of NPF such as local anthropogenic pollution and land ecosystems in introduction.

Page 3, Line 77: "At continental South Pole NPF event are commonly associated with the local anthropogenic pollution during calm weather conditions (Park et al., 2004). In addition, the biomass burning aerosol from South American continental outflow has been observed at Troll Research Station (Fiebig et al., 2009). During the daytime, higher radiation enhances photo-active emissions from land ecosystems (mosses, grasses, and lichens) of the Antarctic Peninsula and can lead to NPF and aerosol growth (Decesari et al., 2020; Quéléver et al., 2022; Schmale et al., 2013). However, land sources are rather unlikely due to a small footprint of emerging land and the associated short overpass over the sparse vegetation."

RC3 (**§1, L.57-L.58**) I suggest adding Sipilä et al. (2016, https://doi.org/10.1038/nature19314) as a key reference for showing the role of Iodic acid in NPF in Marine & Polar environments.

Response: Thank you very much for the information. We added the reference as given below.

Page 3, Line 69: "Indeed, Sipilä et al. (2016) measured iodic acid ($HIO_3$) in Antarctica and found that the Antarctic oceanic regions may be strong sources of molecular iodine, which is then converted to $HIO_3$ in gas-phase reactions."

RC4 (**§1, L.71-L.75**) Let me bring to your attention the publication of Quéléver et al. (2022, https://doi.org/10.5194/acp-22-8417-2022) reporting new particle formation at the Marambio station during the early 2018 (incl. measurement period of the present study), also reporting particle characterization based on size distribution. This reference could also bring context in the discussion presented later in (§3.3.1 and §3.3.2) regarding the relationship between NPF occurrence and meteorological parameters and potential (chemical) source of NPF.

Response: The reference (Quéléver et al., 2022) was addressed in the introduction and discussion sections.

Page 4, Line 90: "To date, number size distribution of particles > 3 nm has been reported by Asmi et al. (2010) at Aboa during from December 29, 2006 to January 29, 2007; by Pant at al. (2011) at Maitri from January 1 to February 28, 2015; by Weller et al. (2015) at Neumayer from January 20 to March 26, 2012; by Jokinen et al. (2018) at Aboa from November 2014 to February 2015; by Weller et al. (2018) at Kohnen during January 2015 and 2016; by Quéléver et al. (2022) at Marambio during the austral summer between January 15 and February 25, 2018; and by Brean et al. (2021) during the PI-ICE cruise from January 25 to February 4, 2019."

Page 12, Line 308: "Recent studies concluded that Antarctic NPF occurred under combined high solar radiation, high temperature and low RH conditions, similar to previous study measured at the Marambio Antarctic research station (Quéléver et al., 2022). Quéléver et al. (2022) found all NPF events were observed during the daytime with high solar radiation (clear-sky conditions), mostly with above-freezing temperature and with low RH."

Page 18, Line 450: "The FR, GR, and CS values agreed well with those reported in previous studies at other Antarctic sites (Järvinen et al., 2013; Kim et al., 2019; Kyrö et al., 2013; Weller et al., 2015), but significantly lower than the values reported by the Quéléver et al. (2022), who showed the average FR and GR were 0.686 $cm^{-3}$ $s^{-1}$ and 4.2 nm $h^{-1}$, respectively."

RC5 (**§2.1, L.107-L.108**) Please clarify for each CPCs if this corresponds to measurement incl. ultrafine particle (corresponding data CN2.5) or particles larger than 10 nm (corresponding data CN10), if mentioned already in the method, there is then no need to specify it later (e.g., L.233 – L.234 or in the caption of Fig.2).

Response: To clarify this issue, we explained for each CPCs in experimental methods and removed the information later.

Page 5, Line 133: "Two condensation particle counters (TSI model 3776 CPC and TSI model 3772 CPC) were used to measure the total number concentration of particles larger than 2.5 (corresponding data $CN_{2.5}$) and 10 nm (corresponding data $CN_{10}$) every 1 s, respectively."

RC6 (**§2.2, L.155-L.159 & L.165-L.168**) Could you explain the reason for the BC comparison between North Atlantic Ocean and Southern Ocean (where northern hemisphere is usually more influence by anthropogenic factors)? The environment being significantly different, I would suggest to clarify the context or to revise the relevance of this additional information to the manuscript.

Response: We agree with the reviewer's comment. Since environmental conditions could be significantly different, the statement on the BC comparison between North Atlantic Ocean and Southern Ocean was removed. We newly added the information about BC concentration measured in Antarctic regions as given below.

Page 8, Line 187: "Hara et al. (2019) measured BC concentration at Syowa station Antarctica from February 2005 until December 2016. They found that the daily median BC concentrations were below the detection limit (0.2 ng $m^{-3}$) to 63.8 ng $m^{-3}$ at Syowa Station (median, 1.8 ng $m^{-3}$; mean, 2.7 ng $m^{-3}$ during the measurement period). During the ACE-SPACE expedition, BC concentration reach its background levels of 19.2 ng $m^{-3}$ (Schmale et al., 2019). Arctic shipborne-observations measured BC concentration throughout the Arctic Ocean and Pacific Ocean during the summer of 2017, all pointing to pristine clean marine air masses with BC values of approximately $20 \pm 10$ ng $m^{-3}$. (Park et al., 2020)."

RC7 (**§3.1**) I would suggest to re-assess the structure of the subsections within 3.1 in order the easily follow the story line of the analysis by, for example, adding a subsection for the meteorological parameters influencing the aerosol particles and their formation (L. 248).

Response: Based on the reviewer's comments, we added a subsection for the meteorological parameters influencing the aerosol particles (3.1.2. Influence of meteorological parameters on NPF events).

RC8 (**§3.1.1, Table 1 & L.230**) Please reformulate the caption for Table 1: e.g., "Monthly median for total particle number concentration >10 nm (CN10) ….". I also suggest to add, in the caption, that the data are filter for pristine / clean conditions only (i.e., data when BC <50 ng.m-3). Finally, review the sentence **L.230** accordingly as **Table 1** does not show the time series for one-hour average but it recaps monthly median values for the year 2018.

Response: We reformulated and modified the caption of Table 1. Also, to clarify this, we rewrote the statement as given below.

Page 10, Line 263: "In addition, monthly medians for total number concentration of particles, size-segregated particles number concentration, CCN number concentration at supersaturation of 0.4%, and metrological parameters are included in Table 1."

RC9 (**§3.1.1, Figure 2**) CN2.5 is visible only from Jan. to Mar., it would be worth to comment on that in the text as well. Also consider to use color set that are color blindless-friendly (e.g., other than green and red in the same plot). (**L.244-L.247**) The correlation analysis CN10 vs NNUC could be more relevant with a bigger data set rather than with monthly averaged values, why not using the one-hour data?

Response: Thank you for pointing this out. We only measured CN 2.5 from January to March due to the instrumental malfunctions. We added the information in the caption of Figure 2 as follows. According to the reviewer's comment, the color of the coastal samples on Figure 1 has been changed. The correlation between $CN_{10}$ and $N_{NUC}$ was calculated by using hourly data and the sentence was modified.

Page 35 (caption of Figure 2): "CN 2.5 data are only available from January to March due to the instrumental malfunctions."

Page 11, Line 276: "Furthermore, the hourly average $CN_{10}$ value was positively correlated with the hourly average $N_{NUC}$ ($R = 0.88$; not shown), implying that the summer maximum of total particle number concentrations was largely influenced by newly formed particles in the Antarctic atmosphere."

RC10 (**§3.1.1, L.270-L.272**) Although the authors focus first on specific meteorological parameters influencing NPF, I suggest to strongly insist on the combination high PAR, high temperatures, low RH altogether rather than finding explanation with one parameter alone. Furthermore, high windspeed would enhance mixing of the emitted compounds and accelerate transport (possibly from further away). Please add reference for enhanced VOC in high windspeed conditions.

Response: As suggested by the reviewer, we added the reference for enhanced VOC in high wind speed conditions. The sentence for enhanced VOC in high wind speed conditions was rewritten accordingly.

Page 12, Line 308: "Recent studies concluded that Antarctic NPF occurred under combined high solar radiation, high temperature and low RH conditions, similar to previous study measured at the Marambio Antarctic research station (Quéléver et al., 2022)."

Page 12, Line 303: "A possible explanation for the wind speed independence is that an increase in wind speed contributes to the increase of cluster size ion number concentrations by friction processes (Virkkula et al., 2007), but it was also accompanied by cloudy conditions."

RC11 (**§3.1.2, L.276 -**) The section 3.1.2 depicts the statistics on the observed NPF events. I would suggest to first mentions the numbers of event observed with a brief description of representative event types (if such grouping can be done), e.g., burst events, nucleation transported + local growth, etc.… incl. example with size distribution surface plot.

Response: We described two types of NPF events including the contour plots of the size distributions as follows.

Page 13, Line 323: "The NPF events were classified into: (1) burst event and (2) nucleation with growth event according to the classification by Dal Maso et al. (2005) as seen in Figure S3. The burst events and nucleation with growth events were observed on 1 January 2018 and 16 December 2018, respectively."

[Figure]

Figure S3. Example of two types of the NPF based on the SMPS data. (a) burst event (1 January 2018) and (b) nucleation with growth (16 December 2018).

RC12 (**§3.1.3, L.301 -**) The start of the section reintroduces the NPF / nucleation presented in the earlier section, I would recommend restructuring the section 3.1. in order to follow a coherent path on the descriptions of NPF events, without reintroducing NPF observation on every subsection.

Response: Based on the reviewer's comment, we restructured the section 3.1. To avoid the description of NPF events repeatedly, we changed the order of section 3.1.3 and 3.1.4 and also modified the sentence.

RC13 (§3.2) For each case study, the manuscript shows the result of CCN data for one supersaturation ratio only, could the authors develop on the reason for using this data only rather than comparing with the information brought with the other super saturation ratio. RCx (**§3.3.2, L.426 -**) Here as well, I suggest adding Sipilä et al. (2016, https://doi.org/10.1038/nature19314) as a key reference for showing the role of Iodic acid in

NPF in Marine & Polar environments.

Response: To clarify this issue, we included the CCN concentration at five different supersaturations for each case study (Figure 6-8). Also, we added the time series of daily mean CCN concentration under five different supersaturations (Figure S9). As suggested by reviewer, we added the reference as well.

[Figure]

Figure 6. Marine NPF event observed from December 9–15, 2018. From top to bottom, the plots are as follows: meteorological variables, the residence time of air masses that passed over the ocean, sea ice and land areas; number size distribution with the standard-SMPS and nano-SMPS, and CCN number concentration. The x-axis represents local time.

[Figure]

Figure 7. Sea ice NPF event observed from January 2–3, 2018. From top to bottom, the plots are as follows: meteorological variables, the residence time of air masses that passed over the ocean, sea ice and land areas; number size distribution with the standard-SMPS and nano-SMPS, and CCN number concentration. The x-axis represents local time.

[Figure]

Figure 8. Multiple NPF event observed from November 16–17, 2018. From top to bottom, the plots are as follows: meteorological variables, the residence time of air masses that passed over the ocean, sea ice and land areas; number size distribution with the standard-SMPS and nano-SMPS, and CCN number concentration. The x-axis represents local time.

[Figure]

Figure S9. Time series of daily mean CCN concentration under different supersaturation conditions.

RC14 (**§3.3.2, L.452 -**) The presence of Penguins close (~2 km) to the measurement location is a determinant parameter impacting the frequency of NPF. Earlier in the section, the authors

discussed extensively on the air mass origin, however this now bring a strong local source for chemicals bases such as ammonia that can trigger NPF. This, however, could only be validated by direct measurement of precursor gases. Hence, I highly recommend discuss this further, as well as the need for gas phase measurement. As such, a close emission point will likely interact with the result interpretation incl. from most / all the air mass origin. Furthermore, if pinguins / birds are only present in the surrounding seasonally, I would further suggest a comparison on the frequency / intensity of the NPF observed between e.g., breading season and start of the spring.

Response: This is a similar question to the one above. Again, we did not measure the precursor gases. Out of 97 observed NPF events, 2 NPF events (4 February 2018 and 18 February 2018) were associated with local fauna. We presented the contour plots of the size distributions and wind roses when predominant wind possibly passing over a penguin colony (around 2 km away from our observation site) in Figure S8. We discussed the influence of local fauna such as penguin colonies on NPF event in the manuscript (Page 20 and Line 513) and the importance of measurements of precursor gases (Page 24 and Line 620).

Page 24, Line 620: "However, further detailed measurements of the chemical properties of aerosol particles and precursor gases (e.g., ammonia) during NPF events are required to better understand the contribution of these compounds to the formation and growth of aerosol particles and to explore their impacts on CCN formation in the remote Antarctic environment."

RC15 (**§3.3.3, L.469 -**) Could the authors bring clarity on the connection between the size distribution data and the CCN data. It is a critical point of this study, as NPF and CCN do not occur at the same time, it would be interesting to account for parameters associated with CCN formation in connection to NPF (particle number/diameter, sinks, survival probability …) Have the authors considered CCN transported from another source (i.e., primary particles)? Alternatively, the authors could present the result given with supersaturation ratio and discuss the link with hygroscopicity (CCN data) and growing particles (DMPS data). I then suggest referring to Chang et al. (2022, https://doi.org/10.5194/acp-22-8059-2022) to complement the analysis.

Response: In the present study, we did not consider the contribution of sea spray aerosols to CCN formation. In the future, we will investigate the physical properties of primary aerosols in Antarctic peninsula and its contribution to CCN using the size distribution data obtained up to coarse mode particles as well as chemical composition information. To clarify the connection between growing particles and CCN, we determined the increase in CCN concentration during growth time (i.e., growth to lower than 40 nm and growth to larger than 40 nm) compared with baseline values at different supersaturations, based on the method by Chang et al. (2022), as shown in Figure 10.

**Technical comments (rmc):**

TC1 (**Title**) Suggestion to revise the title by replacing the preposition "at" by "in the Antarctic Peninsula".

Response: It was corrected.

TC2 (**Abstract, L.24-L.27**) Suggestion to reformulate as there is no direct measurement of

DMS/DMSP. → E.g. "Our estimation of DMPS concentration from satellite chlorophyl data suggest that product of biogenic precursor could be a component of marine NPF, …".

Response: We have corrected this in the abstract as follows:

Page 1, Line 27: "Satellite-estimates for sea surface dimethylsulfoniopropionate (DMSP; a precursor of gaseous dimethyl sulfide) data showed that the production of oceanic biogenic precursors could be a key component in marine NPF events."

TC3 (**§2.1, L.129-L.130**) CCNC supersaturations either 20%, 40%, 60%, 80% 100% or supersaturation ratio of 0.2, 0.4, 0.6, 0.8, 1.

Response: It was corrected.

TC4 (**§3.1.3**) "size spectra" → "size distribution spectra"

Response: It was corrected.

TC5 (**§3.1.3, L.390**) "evets" → "events"

Response: It was corrected. Thank you for finding this error.

TC6 (**§3.3.2, L.413 & L.418**) Please consider the product of oxidation of DMS are those responsible for the NPF. Condensable vapors: Sulfuric acid and Methane sulfonic acid.

Response: We added the information (Page 19 and Line 474).

**Newly added references**

[revised manuscript text omitted]

---

## Author Comment (AC2)

**Reviewer #2: The paper "new particle formation leads to CCN at the Antarctic Peninsula" aim to present one year of data (2018) of SMPS data and to connect it with CCN data.**

I am not sure if there is a contribution at this stage. Similar and much in depth results were published from Jang et al 2019 and Kim et al 2019 on part A part B of ACP. This paper stresses the importance of the ocean, but the air mass classification and the SMPS data analysis is too weak to draw any further conclusion, I am not sure if this paper add anything new to the literature for a high impact journal like ACP.

We would like to thank the reviewer for valuable and constructive comments. Below is our point by point response to each of the comments.

In this study, we simultaneously presented one-year nano SMPS, standard SMPS, and CCN data during the pristine and clean periods (BC concentrations less than 50 ng m$^{-3}$), which enables to suggest more robust and topical hypothesis compared with the previous works by Jang et al. (2019) and Kim et al. (2019). It enables us to improve our understanding NPF and growth events, significantly enhancing CCN concentration. In fact, we newly suggested that spatial scale of NPF around Antarctic peninsula was found to be about 155 km away by using the nano SMPS data and 86% of NPF events were characterized by the simultaneous increase in the CCN concentration by 44% (by median) in the following 8 hours (by median) by comparing both SMPS and CCN data for each NPF event.

The paper also suffers from a poor literature review, totally unbalanced - it somehow reminds controversial marine organic aerosol topics discussed twenty years ago, with few respected scientists arguing over POC-DOC and the effect of organics on the sea spray production (still not solved!).

This comment seems to be not relevant to our topic. The POC-DOC and the effect of organics on the primary sea spray production are not discussed in this manuscript. Authors acknowledge the points raised by the referee and series of references were added in this revised version in the introduction and discussion parts.

(1) Introduction - poorly presented, it cites one paper (Kyro et al 2013) stressing the importance of Antarctic melt ponds waters, purely speculative and not shown in any data, only suggested as a possibility - remote, I would say; if you have an idea of the overall geography of the Antarctic continent and marine surrounding. The paper continue to report papers about animals and emissions, although from studies run very close to emission sources (indeed the island is called Bird island).

Response: In order to address the sources emitted from the animals, we newly included 2 NPF events (4 February 2018 and 18 February 2018) which were associated with local fauna. We presented the contour plots of the size distributions and wind roses when predominant wind possibly passing over a penguin colony (around 2 km away from our observation site) in supporting information. We discussed the influence of penguin colonies on NPF event and also added the references as given below.

Page 20, Line 513: "In fact, 2 NPF events (4 February 2018 for marine air mass origin and 18 February 2018 for multiple air mass origin) were observed when winds were seen to originate

from the south sector where strong emission from the penguin colonies (southeast sector of 106–140°). Figure S8 showed the contour plots of the size distributions and wind roses during those days. Although we did not directly measure the precursor gases such as ammonia and amine that can trigger the NPF, we can speculate that the fauna on the land or at the shore such as penguin and seabird colonies could not be excluded as the potential source of NPF events locally although highly productive and ice melting Weddell sea is coinciding with southeast direction too. Previous studies reported that precursor gases for NPF (e.g., ammonia) can originate from the decomposition of excreta from seabirds and penguins (Lachlan-Cope et al., 2020; Legrand et al., 1998; Liu et al., 2018; Schmale et al., 2013). More recently, Quéléver et al. (2022) proposed that nitrogen-containing species could be land-sourced (e.g., from a high penguin population during the summertime) or marine-sourced (e.g., from the biological activity of plankton in the ocean and melting sea ice). The ammonia from seabird-colony guano is a key factor contributing to bursts of newly formed particles, which are observed in the summertime Arctic (Croft et al., 2016)."

[Figure]

Figure S8. (a) Contour plots of the size distributions and (b) wind rose on 4 February 2018 and (c) contour plots of the size distributions and (d) wind rose on 18 February 2018. The southeast direction (106–140°) is designated as a sector where strong emission from the penguin colonies may originate. The x-axis represents local time.

What is even funnier is that the authors do not mention any of recent open ocean and coastal Antarctic expeditions (i.e. PEGASO, ACE Schmale Baccarini et al, PI-ICE) - that is funny, cause some of the authors were even on board of such cruises (and continue publishing data - without the PIs involved - from such cruises). Overall it is a pity in 2023 these things are still happening - I suggest to write a better and more fair introduction. Also, recent papers published by Australian groups talking about Antarctica and CCN may be worth mentioning and consider in the discussion. At this stage the paper is about a report of measurements.

Response: In this study, we measured the NPF event and CCN activity at land-base station, so we focused primarily on mentions of the previous studies conducted at Antarctic land-base stations for comparison. Based on the review's comment, we newly added and discussed several references for recent open ocean and coastal Antarctic expeditions (Brean et al., 2021; Dall'Osto et al., 2017; Dall'Osto et al., 2022; Decesari et al., 2020; Humphries et al., 2021; Humphries et al., 2023; Schmale et al., 2019; Simmons et al., 2021; Walton and Thomas, 2018) in introduction and discussion sections as follows.

Page 2, Line 51: "Furthermore, open ocean and coastal Antarctic expeditions such as SIPEXII (Sea Ice Physics and Ecosystems eXperiment, 2012; Humphries et al., 2015; Humphries et al., 2016), PEGASO (Plankton-derived Emissions of trace Gases and Aerosols in the Southern Ocean, 2015; Dall'Osto et al., 2017; Decesari et al., 2020; Fossum et al., 2018), ACE-SPACE (Antarctic Circumnavigation Expedition – Study of Preindustrial-like Aerosol Climate Effects, 2017; Schmale et al., 2019; Walton and Thomas, 2018), PCAN (Polar Cell Aerosol Nucleation, 2017; Simmons et al., 2021); PI-ICE (Polar atmosphere-ice-ocean Interactions: Impact on Climate and Ecology, 2019; Brean et al., 2021; Dall'Osto et al., 2022) studies on the influences of marine aerosols on climate and ecology."

Page 4, Line 90: "To date, number size distribution of particles > 3 nm has been reported by Asmi et al. (2010) at Aboa during from December 29, 2006 to January 29, 2007; by Pant at al. (2011) at Maitri from January 1 to February 28, 2015; by Weller et al. (2015) at Neumayer from January 20 to March 26, 2012; by Jokinen et al. (2018) at Aboa from November 2014 to February 2015; by Weller et al. (2018) at Kohnen during January 2015 and 2016; by Quéléver et al. (2022) at Marambio during the austral summer between January 15 and February 25, 2018; and by Brean et al. (2021) during the PI-ICE cruise from January 25 to February 4, 2019."

Page 4, Line 105: "Ship-based observations during the ACE-SPACE found that the fraction of particle serving as CCN was higher near the coast of Antarctica compared to open ocean, resulting from multiple processing cycles of dissipating and condensing clouds and/or the higher availability condensable gases originating from marine microbial activity (Schmale et al., 2019)."

Page 18, Line 458: "In comparison, Jokinen et al. (2018) reported that GR values ranged from 0.3 to 1.3 nm h$^{-1}$ at Aboa, and Brean et al. (2021) showed GR of 0.4 to 0.6 nm h$^{-1}$ measured during the PI-ICE cruise."

Page 21, Line 537: "The values are in line with previous studies published from the Antarctic regions. Humphries et al. (2023) reported CCN concentrations nearby East Antarctic observations from Macquarie Island and Kennaook / Cape Grim as well as recent ship voyages of the RSV Aurora *Australis* and the RV *Investigator* in the region. The median CCN value at a SS of 0.5% was in the ranges of 88–145 cm$^{-3}$ at Macquarie Island, 57–158 cm$^{-3}$ at Kennaook / Cape Grim, and 40–230 cm$^{-3}$ during the voyages (No voyage data exist for the winter months), respectively. The PCAN project exhibited that a median particle number concentration larger than 3 nm of 354 cm$^{-3}$ was observed from the voyage and median CCN at 0.55 % supersaturation were 167 cm$^{-3}$, implying approximately half the particles measured as CN$_3$ could be activated as CCN (Simmons et al., 2021). Recently, several ship-based measurements over the Southern Ocean found significantly increased MSA concentrations in air masses originating close to the Antarctic coastline, alongside enhancements in CCN concentration (Humphries et al., 2021)."

(2) BC data. Not sure if this is all necessary - if you want to compare other data, consider to compare Antarctic station or Arctic station, mentioning Mace Head is a "little" bit out of scope here. It is evident that marine Atlantic aerosols has nothing to do with Antarctic - consider remove all this section for eventually a future publication elsewhere.

Response: We agree with the reviewer's comment. Since environmental conditions could be significantly different, the statement on the BC comparison between North Atlantic Ocean and Southern Ocean was removed. We newly added the information about BC concentration measured in Antarctic regions as given below.

Page 8, Line 187: "Hara et al. (2019) measured BC concentration at Syowa station Antarctica from February 2005 until December 2016. They found that the daily median BC concentrations were below the detection limit (0.2 ng m$^{-3}$) to 63.8 ng m$^{-3}$ at Syowa Station (median, 1.8 ng m$^{-3}$; mean, 2.7 ng m$^{-3}$ during the measurement period). During the ACE-SPACE expedition, BC concentration reach its background levels of 19.2 ng m$^{-3}$ (Schmale et al., 2019). Arctic shipborne-observations measured BC concentration throughout the Arctic Ocean and Pacific Ocean during the summer of 2017, all pointing to pristine clean marine air masses with BC values of approximately $20 \pm 10$ ng m$^{-3}$ (Park et al., 2020)."

(3) Definition of different events - beside the dozens of papers discussing different types of nucleation with K-means clustering, do not you have better examples than the ones presented? Out of the 97 NPF events, you may want to have a look at what you presented:

Response: We did not analyze K-means clustering. Most of papers discussing different types of NPF based on the K-means clustering focused on the characteristics of NPF event. In this work, our main aim is to investigate the connection between NPF and CCN for one-year dataset. Instead, we added more examples for NPF event in Figure S3 and Figure S8. Figure S3 included examples of two types of the NPF event based on the SMPS data such as burst event and nucleation with growth. Figure S8 showed examples of the NPF event when winds were seen to originate from the south sector with potentially strong emissions from the penguin colonies.

Figure 6 - is this a NPF event or did you by mistake add a wrong figure? Given you comment it in the text, I assume this is the right figure. I am not sure this is a NPF event, it looks to me it is a background mode of about 20nm that is existing in a large area, it reminds what was discussed in O´Dowd et al (GRL, 2010) and in other papers of open ocean slow growing ultrafine particles. What I am not sure is that if there particles are growing in Figure 6, it looks to me they stay there, and they were detected as a background small Aikten (20-30nm) mode - could be primary or secondary or something interesting, surely not a "tyipcal" NPF for a marine case I would say.

Response: O'Dowd et al. (2010) presented open ocean new particle production and growth events occurred during periods of high oceanic productivity over the Northeast Atlantic. As mentioned by reviewer for BC issue, since environmental conditions between Northeast Atlantic Ocean and Southern Ocean are significantly different, we cannot compare the NPF event observed between both areas. In the current study, we cannot find particle formation starting directly from the lower end of the particle size spectrum (2.5 nm). The initial diameter of particles that arrived to the measurement site during the NPF ranged from 4 nm to 16 nm, indicating the spatial extension of regional NPF event. Our results are broadly in line with

previous results published from the Arctic and Antarctic regions. We discussed this issue as given below.

Page 13, Line 341: "A ship-borne field campaign over Arctic Ocean found a trimodal distribution at $18 \pm 3$ nm, $53 \pm 6$ nm and $150 \pm 6$ nm for open-ocean marine Arctic NPF event and a bimodal distribution at $24 \pm 3$ nm and $151 \pm 3$ nm for Open-ocean terrestrial Arctic NPF event (Park et al., 2020). Lachlan-Cop et al. (2020) presented k-mean cluster analysis of particle size distribution measured at Halley, Antarctica, showing a nucleation peak at 15 nm for "nucleation" ultrafine category and a nucleation peak at 27 nm for "bursting" ultrafine category."

Figure 7 The second may be a NPF event, the first one is a little bit a burst, of 30 min, followed by a 20-30nm mode lasting for several hours - I suggest to look for better examples or to use a more detailed classication.

Response: Based on the reviewer's comments, we changed the example of sea-ice NPF event as shown in Figure 7. The sentence was modified as given below.

Page 16, Line 406: "The NPF event with subsequent particle growth were detected from around 19:00 on January 13, 2018, to around 08:00 on January 14, 2018 (Figure 7). Air temperature and RH during the event were 0.1 °C and 85%, respectively, while solar radiation decreased from 131.7 to 0.2 W $m^{-2}$. Winds were mild and stable (1.9–5.7 m $sec^{-1}$), with a prevailing northwesterly (262–350°) direction and air masses predominantly coming from sea-ice. The average retention times of the 2 d back trajectories traveling over ocean, sea-ice, and land were 20.0, 20.9, and 7.1 h, respectively, indicating sea-ice-influenced air masses (Figure 5b). During the NPF event, both total DMSP and chlorophyll exposure values are stable, with median exposures of 13.3 nmol $L^{-1}$ and 0.2 mg $m^{-3}$, respectively.
During the event, $CN_{2.5}$ and $CN_{10}$ increased to 5669 and 5097 $cm^{-3}$, respectively. Furthermore, the median $N_{NUC}$, $N_{AIT}$, and $N_{ACC}$ values were 508, 376, and 66 $cm^{-3}$, respectively. Elevated CCN concentrations at 0.2 and 0.4 % supersaturations were not observed, whereas CCN concentrations at 0.6, 0.8, and 1.0 % supersaturations slightly increased during the event. For instance, CCN concentration at 0.8 % supersaturation was 517 $cm^{-3}$ at 20:00 on January 13, then increased to 688 $cm^{-3}$, until 23:00 on January 13. The CCN concentration at 0.6, 0.8, and 1.0% supersaturations increased by 11%, 33%, and 58%, respectively."

[Figure]

Figure 7. Sea ice NPF event observed from January 13–14, 2018. From top to bottom, the parameters are as follows: meteorological variables, the residence time of air masses that passed over the ocean, sea ice and land areas, number size distribution with the standard-SMPS and nano-SMPS, and CCN number concentration. The x-axis represents local time.

Figure 8 are these data in local time? Please define the time, it looks this is a night time nucleation event?

Response: We used the local time (LST). As suggested by the reviewer, we define the time in the figure captions. As illustrated in Figure 8, initial stages of nucleation are observed from 17:00 when sunlight exists. Thus, the plausible explanation for the NPF event is that the actual formation and growth occurred during daylight hours upwind from measurement location, but very slow growth continued over the Antarctic Peninsula allowing the detection of ～7 nm particles after the sunset.

(4) classification of air masses. Please consider to discuss different types of environment and to discuss it well, especially cause there are marine simpagic and pelagic zones, including the consolidated pack sea ice and the sea ice marginal zone. All these environments are more complex than a simple ocean - sea ice area. I suggest a better analysis, perhaps using a clustering method such us the one presented in Jang 2022 (Science of the Total Environment 803 (2022) 150002). The study of Jang et al 2022 also stress that EAP and WAP are main sources of ultrafine particles (10-25nm) so in contrast to what this current study suggests.

Response: In the present study, we classified air mass origins into ocean, sea-ice (including marginal ice zone and pack-ice area), and land. Since we only refer to the air mass origins when

NPF event occurred, we did not need to perform the clustering analysis. In addition, we did not divide sea-ice zone into marginal ice zone and pack-ice area, because most of NPF events (82%) were observed in ocean-influenced air masses and remaining 18% of NPF event were influenced by sea-ice or land. Jang et al. (2022) considered the domains of pelagic (open ocean area), first-year ice, and multi-year ice zones during entire study periods (from December 2018 to April 2019 and from November 2019 to February 2020). Moreover, Jang et al. (2022) defined the ice zone as maximum sea ice extent observed in August 2018 and 2019. The clusters of air masses were reclassified into three groups representing the Antarctic open ocean (AOO) ($n$ = 1983), western Antarctic Peninsula (WAP) ($n$ = 1675), and eastern Antarctic Peninsula (EAP) ($n$ = 1356). Then, they showed the hourly mean number concentration of newly formed particles ($CN_{10-25}$) assigned for the AOO, WAP, and EAP. Therefore, a role of first-year sea ice on the particle concentrations was addressed in the previous work (Jang et al., 2022) and was out of scope of this study.

The current paper aims to give a clear message: it is all marine open ocean and little sea ice. This is in contrast with the literature, and recent studies (Jang 2022). Unfortunately the previous studies, Jang 2019 part 1 and part 2 did not consider sea ice air masses, and only partially presented data showing only data from open ocean seas from the north. I suggest a more in depth analysis, at the moment this is a qualitative analysis showing some very broad classifications of both air masses and NPF events with a broad terminology - broadly concluding the identical results published before from a much bigger dataset (2009 till 2016) with both SMPS and CCN data.

Response: This is a similar question to the above. Again, the definition of air mass origins is different. Jang et al. (2022) compared the mean value of newly formed particles ($CN_{10-25}$) for each air mass during entire sampling periods, whereas we compared the NPF properties according to air mass origin when NPF event occurred. In the present study, we focused on the NPF characteristics and its linkage with CCN properties. To clarify the connection between growing particles and CCN, we determined the increase in the median CCN concentration at different SS during growth larger than 40 nm and growth smaller than 40 nm times compared with background times based on the method by Chang et al. (2022). Detailed information is provided in the manuscript as given below.

Page 22, Line 574: "To understand the contribution of growing particles on the CCN concentrations during NPF event, we determined the increase in CCN concentration during growth periods (i.e., growth to smaller than 40 nm particles and growth to larger than 40 nm particles) compared to baseline values (black) under different supersaturation conditions (Figure 10), according to the method suggested by Chang et al. (2022). When particle growth was smaller than to 40 nm (growth ≤ 40 nm), the mean CCN concentrations increased by 59–178 $cm^{-3}$ for a SS of 0.2 %–1.0 %, representing a 172.3–216.7 % increase compared to the values during baseline conditions. When particle growth was larger than to 40 nm (growth > 40 nm), the mean CCN concentrations increased by 57–227 $cm^{-3}$ for a SS of 0.2 %–1.0 %, representing a 169.9–249.1 % increase compared to baseline values. Our results indicate that particles formed from NPF events can lead to the significantly enhanced CCN concentration in Antarctic Peninsula, and this effect is more pronounced if we consider particle growth larger than 40 nm, consistent with ship-based observations (Chang et al., 2022) and aircraft-based observation (Willis et al., 2016) in the Canadian Arctic during summer."

[Figure]

Figure 10. The increase in CCN concentration during growth to larger than 40 nm particles (green) and smaller than 40 nm (blue) times compared with background times at five different supersaturations.

(5) Wind roses or potential source function analysis to prove and or exclude local sources as suggested by reviewer 1.

Response: We agree with the reviewer's comment. To minimize the influence of local pollution sources from a power generator and crematory during the data analysis, we used black carbon concentration, wind speed and wind direction data as described in section of 2.2. Thus, the northeastern direction of 355–55° is designated as a local pollution sector due to emissions from the power generator and crematory. Data collected from this sector were absolutely discarded. Again, to address the potential local source from wild-life emissions, we included wind roses in Figure S8 as mentioned above.

 I suggest to publish this paper in a lower impact journal or in measurement report.

Response: Authors do believe the revised version brings significant scientific advancement to the topical area.

**Newly added references**

[revised manuscript text omitted]

---

## Author Comment (AC3)

**Reviewer #3:**

Interesting paper, the authors present data about ultrafine particles from the Antarctic peninsula respectively the South Shetland Islands. There and at the tip of the peninsula at Grahamland, several of the Antarctic research stations are located, an on the first glance good location to investigate extremely clean air.

The authors present ultrafine particle measurements in the size range of about 8-10 nm, the threshold for nanoparticles typically for nucleation, to about 30 - 60 nm and claim that these particles contribute significantly to cloud condensation nuclei. This statement concerning CCN is well accepted and important for the local meteorology and climate.

We thank the reviewer for providing valuable suggestions that improved the readability of our revised manuscript.

However, the attribution of the observed particles to new particle formation (NPF), respectively gas to particle conversion from natural particle precursors is not supported by the data presented. Gas to particle conversion from biogenic emissions and DMS related sulphur compounds first leads to particles in the nucleation mode below 10 nm and growth to the measured sizes would need several hours (Kulmala et al, 2013). Such particles were not observed although the instrumentation used was specially included to investigate the particle nucleation size range down to 2.5 nm, an observation that is even stated by the authors. The particles were, accordingly, most likely produced elsewhere and advected to the site, as also stated in the text. Where and by which process are the particles produced?

Response: It should be stressed that particles at 2.5nm or cluster sizes are observed only at locations were the formation process takes place. If the particles were detected at larger sizes, they were advected from formation region elsewhere. Since the spatial scale of NPF (median value: 155 km) was estimated according to the local wind speed and time during which a distinct nucleation mode can be observed at the sampling site, we cannot pinpoint to an accurate location where particles in the nucleation mode below 10 nm were actually produced. Therefore, our estimates are only approximate. Based on the air mass analysis, chlorophyll and DMSP exposure, sea-ice coverage, we speculated that the particles were produced by photooxidation of biogenic DMS for marine NPF event or by halogen compounds released from ice-covered areas for sea-ice NPF event. In addition, terrestrial sources (e.g., animal colonies or vegetation) could have influenced the NPF for multiple NPF event. However, further measurements of the chemical properties of aerosol particles and precursor gases are required to clarify the NPF processes in the future.

The authors present a hypothesis about a potential production process and source location. Such a localized process only in a limited geographic location upwind and is, however, not supported by corresponding data on the regional distribution of nucleation precursors.

Response: To clarify the potential local source (e.g., emission from the animals), we included wind roses in Figure S8 as mentioned above. 2 NPF events (4 February 2018 for marine air mass origin and 18 February 2018 for multiple air mass origin) were observed when winds were seen to originate from the south sector where strong emission from the penguin colonies. Figure S8 showed the contour plots of the size distributions and wind roses during those days. However, given the proximity of penguin colony (~2-3 km away) we should have observed

newly formed particles in bellow 10 nm sizes. That was not case which strongly suggests that the nucleation event took place much further upwind and air mass overpass over the penguin colony most likely contributed to growth of existing particles without forming new.

Page 20, Line 513: "In fact, 2 NPF events (4 February 2018 for marine air mass origin and 18 February 2018 for multiple air mass origin) were observed when winds were seen to originate from the south sector where strong emission from the penguin colonies (southeast sector of 106–140°). Figure S8 showed the contour plots of the size distributions and wind roses during those days. Although we did not directly measure the precursor gases such as ammonia and amine that can trigger the NPF, we can speculate that the fauna on the land or at the shore such as penguin and seabird colonies could not be excluded as the potential source of NPF events locally although highly productive and ice melting Weddell sea is coinciding with southeast direction too. Previous studies reported that precursor gases for NPF (e.g., ammonia) can originate from the decomposition of excreta from seabirds and penguins (Lachlan-Cope et al., 2020; Legrand et al., 1998; Liu et al., 2018; Schmale et al., 2013). More recently, Quéléver et al. (2022) proposed that nitrogen-containing species could be land-sourced (e.g., from a high penguin population during the summertime) or marine-sourced (e.g., from the biological activity of plankton in the ocean and melting sea ice). The ammonia from seabird-colony guano is a key factor contributing to bursts of newly formed particles, which are observed in the summertime Arctic (Croft et al., 2016)."

[Figure]

Figure S8. (a) Contour plots of the size distributions and (b) wind rose on 4 February 2018 and (c) contour plots of the size distributions and (d) wind rose on 18 February 2018. The southeast direction (106–140°) is designated as a sector where strong emission from the penguin colonies may originate. The x-axis represents local time.

The size distributions shown are well in agreement with known anthropogenic emissions. Fossil fuel generators are used at all the Antarctic stations, see also the section in the manuscript about the sector which might be polluted by the own power generation. There are several stations about 20 km and several others within ~ 150 km upwind. The plumes of these research

station generators may not be strong enough to produce a measureable signal at neighboring stations, however, they contribute to background levels depending on wind direction and also point towards another and likely stronger anthropogenic source, which appears from time to time at the same locations, in direct vicinity of the research stations in austral summer, always for a couple of days.

Such a source for ultrafine particles are the supply vessels providing support for the stations (Hobbs et al, 2000, Kivekäs et al, 2014, Junkermann and Hacker, 2022). Their plumes are under selected cloud patterns visible from satellite for distances of more than 100 km especially in an otherwise extreme clean environment (Twomey, 1977, Rosenfeld, 2000). Definitely, ships, which produce a much stronger emission than a mid-size research station generator have no fixed location. However, their position, type and size is readily available from AIS marine traffic repository. The vessels normally stay even for a couple of days close to the research stations. Alternatively, also larger commercial cruise vessels appear increasingly in the area. They move slowly or stay locally for several hours to enable tourist excursions. Pictures in Google Earth document all these anthropogenic activities in the area. Also, these vessels are traceable via AIS.

The paper thus indicates a different problem, the pollution of the Antarctic environment by increasing anthropogenic shipping activities on top of unavoidable research station supply. However, the obvious anthropogenic pollution in this pristine location is not discussed in the manuscript.

Response: We agree with the reviewer's comment. Anthropogenic activity and local contamination (from fossil fuel generators) can influence the size distribution of Antarctic aerosol particles, including the NPF events. To minimize the impact of local contamination, we used black carbon (BC) concentration, wind speed and wind direction data as described in section of 2.2. The observatory is located ~400 m southwest of the main station buildings (e.g., a power generator and crematory). Thus, the northeastern direction (355–55°) is designated as a local pollution sector due to emissions from the power generator and crematory. Data collected from this sector were discarded without considerations. In addition, when BC concentration was higher than 50 ng m$^{-3}$, data were also excluded from analysis regardless of wind direction. Median value of BC concentrations during NPF events was 21.0 ng m$^{-3}$. We compared BC concentration for each air mass as shown in Figure S5. We addressed this issue in the manuscript as given below.

Page 7, Line 164: "As the observatory is located ~400 m southwest of the main station buildings and several kilometers away from other research stations, measurement data were impacted by local emissions from station activities (e.g., power generators and incineration) or anthropogenic pollutions near the observatory (e.g., plumes from other research station about several kilometers, vessels providing research station supply, and commercial cruise vessels)"

Page 14, Line 353: "Median BC concentration for marine, sea-ice, and multiple air masses found to be 23.8 ng m$^{-3}$, 12.7 ng m$^{-3}$, 9.8 ng m$^{-3}$, respectively, (Figure S5), indicating pristine clean air masses with minimum influence from anthropogenic pollutions during each NPF event case."

[Figure]

Figure S5. Median, 25 and 75 percentile BC concentration for marine, sea-ice, and multiple air masses analyzed in this study.

Literature:

Twomey, S. The influence of pollution on the shortwave albedo of clouds, J. Atmos. Sci., 34, 1149–1152 (1977)

Rosenfeld D., Suppression of Rain and Snow by Urban and Industrial Air Pollution, Science, 287, 1793 (2000)

Hobbs, P. V., et al. Emissions from Ships with respect to their Effects on Clouds, J. Atmos. Sci., 57 2570-2590 (2000)

Kulmala, M., et al. Direct observations of atmospheric aerosol nucleation, Science, 339, 943–946, https://doi.org/10.1126/science.1227385, (2013).

Kivekäs, N. et al,: Contribution of ship traffic to aerosol particle concentrations downwind of a major shipping lane, Atmos. Chem. Phys., 14, 8255–8267, https://doi.org/10.5194/acp-14-8255-2014, (2014)

Junkermann, W. & Hacker, J., Unprecedented levels of ultrafine particles, major sources, and the hydrological cycle, Nature Scientific Reports, 12:7410 | https://doi.org/10.1038/s41598-022-11500-5, (2022)

**Newly added references**

Croft, B., Wentworth, G. R., Martin, R. V., Leaitch, W. R., Murphy, J. G., Murphy, B. N., Kodros, J. K., Abbatt, J. P. D., and Pierce, J. R.: Contribution of Arctic seabird-colony

ammonia to atmospheric particles and cloud-albedo radiative effect, Nat. Commun., 7, 13444, https://doi.org/10.1038/ncomms13444, 2016.

Legrand, M., Ducroz, F., Wagenbach, D., Mulvaney, R., and Hall, J.: Ammonium in coastal Antarctic aerosol and snow: Role of polar ocean and penguin emissions, J. Geophys. Res., 103, 11043–11056, 1998.

Quéléver, L. L. J., Dada, L., Asmi, E., Lampilahti, J., Chan, T., Ferrara, J. E., Copes, G. E., Pérez-Fogwill, G., Barreira, L., Aurela, M., Worsnop, D. R., Jokinen, T., and Sipilä, M.: Investigation of new particle formation mechanisms and aerosol processes at Marambio Station, Antarctic Peninsula, Atmos. Chem. Phys., 22, 8417–8437, https://doi.org/10.5194/acp-22-8417-2022, 2022.

---

## Author Response (AR2)

Dear Authors,

thanks for the revised version. You have addressed the referee comments very well!

I have few additional minor comments for the manuscript at this stage. Please see the list below.

Tuukka Petäjä

We would like to thank the editor for valuable and constructive comments and suggestions. Our point-by-point responses to the editor's comments have been provided below.
* * *
Editor comments to Park et al. (2023) New particle formation leads to enhanced cloud condensation nuclei concentrations in Antarctic Peninsula

Abstract:

line 23: Regarding the the spatial scale, please shortly describe, how you reacted the value of 155 km for the scale.

Response: As suggested by editor, we described the information on how to estimate the value of 155 km to the abstract.

Page 1, Line 22: "We estimated the spatial scale of NPF by multiplying the time during which a distinct nucleation mode can be observed at the sampling site by the locally measured wind speed."

Sect 2.2:

line 194 onwards. BC concentration represented with a maximum of 6 significant figures. Not realistic. Please correct the mean concentrations and standard deviations to a reasonable accuracy.

Response: We corrected the mean concentrations and standard deviations as follows.

Page 8, Line 195: "Of the total time period assessed, pristine air conditions represented 30% (mean value of BC: $6 \pm 6$ ng m$^{-3}$), clean for 44% (mean value of BC: $30 \pm 10$ ng m$^{-3}$), lightly polluted 19% (mean value of BC: $69 \pm 14$ ng m$^{-3}$), moderately polluted 6% (mean value of BC: $150 \pm 47$ ng m$^{-3}$), polluted 1% (mean value of BC: $499 \pm 174$ ng m$^{-3}$), and extremely polluted less than 1% (mean value of BC: $1537 \pm 595$ ng m$^{-3}$)."

Sect 2.3:

The size distribution is measured until 300 nm. Do you consider the larger sizes to contribute to the CS? How about the supermicron aerosol particles?

Response: We did not consider the contribution of particles larger than 300 nm to the CS. In the future, we will investigate the impact of larger size particles on CS by measuring the number size distribution of supermicron aerosol particles.

Sect 3.2.3:

line 428: ... observed grown mode... Please improve the sentence.

Response: Thank you for pointing this out. Based on the editor's comments, the sentence was rewritten as given below.

Page 17, Line 432: "This suggests that the actual formation and growth occurred during daylight hours upwind from measurement location, but very slow growth continued over the Antarctic Peninsula allowing the detection of observed grown mode at ~ 7 nm after the sunset."

Sect 3.3.1:
line 463 .. were the highest...

Response: Based on the editor's comments, the sentence was modified as given below.

Page 18, Line 469: "There was no difference in the median value in solar radiation, while the median values for chlorophyll exposure and DMSP exposure were higher in air masses originating from the ocean than in air masses originating from the sea ice."

Sect 3.3.2:

line 469: ... were the highest...

Response: Based on the editor's comments, the sentence was modified as given below.

Page 19, Line 477: "The chlorophyll exposure and DMSP exposure during marine NPF events were higher than those during sea ice NPF events"

line 480: Please clarify the sentence starting "Although sea-ice algae...". I don't see the connection between the satellite not being able to detect the biological activity and the fact that the exposure (based on trajectories?) was lower.

Response: Sea ice algae are microscopic algae that grow within and beneath sea ice in polar regions. These organisms are an essential part of the polar marine ecosystem and play a crucial role in the food web. However, satellites are not typically used to directly measure the biomass of sea ice algae because satellite sensors cannot penetrate through thick layers of sea ice to directly measure the biomass of algae beneath it (Lee et al., 2015; Lange et al., 2017). Thus, calculated chlorophyll exposures (i.e., satellite-estimates of biological activity) cannot account for the biological activities thriving within and beneath of the sea ice. To clarify this issue, we added the information in experimental section (section 2.4) and the sentence was modified as given below.

Page 10, Line 253: "However, satellites are not typically used to directly measure the biomass of sea ice algae because satellite sensors cannot penetrate through thick layers of sea ice to directly measure the biomass of algae beneath it (Lee et al., 2015; Lange et al., 2017). Thus, calculated chlorophyll exposures (i.e., satellite-estimates of biological activity) cannot account for the biological activities thriving within and beneath of the sea ice."

Page 19, Line 488: "The air mass exposure to chlorophyll and DMSP for sea-ice NPF events were 1.8 and 2.7 times lower than those of marine NPF events."

line 503: Please clarify the connection between the higher halogen concentrations and the frozen iodine containing solutions. The last sentence is not well connected to the rest of the paragraph.

Response: We agree with editor's comments. The sentence was removed in the manuscript.

line 513: ... sources of precursor gases leading to NPF.

Response: According to the editor's comments, the sentence was improved as given below.

Page 20, Line 519: "seabird colony emissions are the likely sources of precursor gases to NPF (e.g., ammonia and amine) (Quéléver et al., 2022)."

Sect 3.3.3:

line 553: We newly calculated... Please refrase.

Response: We provided percentage values for the increase of CCN concentrations during NPF event and compared them between each air mass origin in Line 555. The CCN increase rate is not relevant to the scope of the present study. Thus, the sentence was removed.

line 555: CCN rate?

Response: As mentioned above, the sentence was removed.

line 578: No need to have the color indicated in the main text.

Response: It was removed.

line 580-583: The percentage with 4 significant figures is too much. Please round the percentages to more reasonable values.

Response: It was corrected.

Table 1: please add the information of the measurement location and the time to the table caption.

Response: We added the information according to the editor's comments.

Table 1 and elsewhere: The SI abbreviation for second is s. Please correct here and throughout the paper

Response: It was corrected throughout the manuscript.

Figures 6,7,8: please label the subplots a,b,c,d, etc and refer in the figure caption.
Selecting a different color palette to the size distribution data would allow better readability.

Response: We added the subplots for the Figures and changed the color palette to the size distribution data.

[Figure]

Figure 6. Marine NPF event observed from December 9–15, 2018. (a–b) meteorological variables, (c) the residence time of air masses that passed over the ocean, sea ice and land areas, (d) total DMSP and chlorophyll exposures, (e–f) number size distribution with the standard-SMPS and nano-SMPS, and (g) CCN number concentration. The x-axis represents local time.

[Figure]

Figure 7. Sea ice NPF event observed from January 13–14, 2018. (a–b) meteorological variables, (c) the residence time of air masses that passed over the ocean, sea ice and land areas, (d) total DMSP and chlorophyll exposures, (e–f) number size distribution with the standard-SMPS and nano-SMPS, and (g) CCN number concentration. The x-axis represents local time.

[Figure]

Figure 8. Multiple NPF event observed from November 16–17, 2018. (a–b) meteorological variables, (c) the residence time of air masses that passed over the ocean, sea ice and land areas, (d) total DMSP and chlorophyll exposures, (e–f) number size distribution with the standard-SMPS and nano-SMPS, and (g) CCN number concentration. The x-axis represents local time.

References

Lee, Y.J., Matrai, P.A., Friedrichs, M.A., Saba, V.S., Antoine, D., Ardyna, M., Asanuma, I., Babin, M., Bélanger, S., and Benoît-Gagné, M.: An assessment of phytoplankton primary productivity in the Arctic Ocean from satellite ocean color/in situ chlorophyll-a based models, J. Geophys. Res., 120, 6508–6541, doi: 10.1002/2015JC011018, 2015.

Lange, B. A., Katlein, C., Castellani, G., Fernández-Méndez, M., Nicolaus, M., Peeken, I., and Flores, H.: Characterizing spatial variability of ice algal chlorophyll a and net primary production between sea ice habitats using horizontal profiling platforms, Front. in Mar. Sci., 4, 349, 2017.